# Exploring structural, functional, evolutionary, and genetic characteristics of sugar transporters in maize and their roles in abiotic stress tolerance

Md. Sohel Mia[1◉], Md Suzauddula[2◉], Tao Yang[3◉], Rui Li[4◉], Fang Li[5], Jianbo Mi[6], Chao Xia[5], Tanveer A. Wani[7], Seema Zargar[8], M. Atikur Rahman[9]*, Md. Mahmudul Hasan[1,3]*

1 Department of Nutrition and Food Technology, Jashore University of Science and Technology, Jashore, Bangladesh, 2 Department of Food, Nutrition, Dietetics and Health, Kansas State University, Manhattan, Kansas, United States of America, 3 Biotechnology and Germplasm Resources Institute, Yunnan Academy of Agricultural Sciences, Kunming, China, 4 College of Agronomy, Academy of Agriculture and Forestry Sciences, Qinghai University, Xining, China, 5 Maize Research Institute, Sichuan Agricultural University, Chengdu, China, 6 Xichang Shijia State-Owned Forest, Liangshan, China, 7 Department of Pharmaceutical Chemistry, College of Pharmacy, King Saud University, Riyadh, Saudi Arabia, 8 Department of Biochemistry, College of Science, King Saud University, Riyadh, Saudi Arabia, 9 Department of Biological Sciences, Alabama State University, Montgomery, AL 36104, USA

◉ These authors contributed equally to this work.
* hasanm_agb@yahoo.com (MMH); mrahman@alasu.edu (MAR)

## Abstract

Sugars are the structural building blocks of carbohydrates, which are transported through a series of transporters in plant. To explore the molecular mechanisms of how transporters, play roles in uptake, transport and mobilization of sugars in maize, a series of bioinformatics analyses were done to identify and characterize the transporters. Following the analyses, 60 sugar transporters were identified in maize, which shared eight (STP, PLT, ERD6, INT, TMT, pGlcT, SUC, and VGT) clades during phylogenetic analysis. Due to having significant differences in molecular weight, multiple beta-strands, transmembrane helices, and 11–12 transmembrane domains, the transports might play significant variations in functional properties. Since most transporters are plasma membrane bound, and have the highest homolog pairs (39) with *S. bicolor* during synteny analysis, the transporters might be involved in intercellular sugar transport that are conserved and significantly duplicated during the process of evolution. The lowest binding affinity (ΔG: −7.1 kcal/mol) in ZmVGT1-Suc docked complex, and most commonly found hydrogen bond mediated attachment of valine residue ligand might represent the complex stability and functional integrity of the complex. Indeed, the RMSD deviation of 1 (for ZmST10-Gal) to 3 Å (for ZmST10-Glu) among the docked complexes might guide the subtle conformational differences that could impact the functional roles of the complexes. Next, during co-expression analysis, clustering of 491 genes with 43 maize sugar transporter into four co-expression clusters and five different metabolic pathways might guide their inter regulatory roles in interacting different metabolic pathways. More specifically,

**Data availability statement:** All data are included in the manuscript.

**Funding:** The Natural Science Foundation of Sichuan Province (Grant No. 2022NSFSC1774), the National Natural Science Foundation of China (Grant No. 32101673), the Liangshan Prefecture Science and Technology Program (24YYYJ0184 and 24YYYJ0183), the Sichuan Science and Technology Program (Grant No. 2021YFYZ0017), and the Academician Expert Workstation of Yunnan Province (to Han Lan, Grant No. 202305AF150052) provided support for this work. The Yunnan Major special research project "Creation and application of special bio-fertilizer for Plateau characteristic economic crops in Yunnan Province" (202202AE090015) also provided money for the work. Ongoing Research Funding Program project number (ORF-2026-357), King Saud University, Riyadh, Saudi Arabia, for providing technical and instrumental support.

**Competing interests:** According to the authors, no known conflicting financial interests or personal ties may have impacted any of the work reported in this research.

co-expression of *ZmSTP9* and *ZmPLT10* with the MYB8 and A6b stress-responsive transcription factors might guide their stress regulatory mechanisms. The RNA-Seq based observation of differential tissue specific expression and expression under salinity, drought, nitrogen deficiency, and heat stress and qRT-PCR mediated validation of differential tissue specific expression and upregulation of *ZmPLT1*, *ZmPLT8*, *ZmSTP1*, *ZmTMT1*, and *ZmSUC3* under salinity stress might guide their potential roles in abiotic stress tolerance. The plasma membrane localized validation of subcellular localization of ZmPLT1 and ZmPLT8 proteins might guide the consistent results between dry and wet lab experiments. Therefore, the identified and characterized maize sugar transporters through integrated dry and wet lab experiments might guide the future research in developing abiotic stress tolerant maize and exploring the molecular mechanism of stress tolerance trough transporter guided regulation of maize abiotic stress signaling pathways following circuit enabled synthetic biology approaches.

## Introduction

Sugars are essential for plant metabolism, serving as the primary energy source and carbon intermediates, and playing a crucial role in signal transduction and stress tolerance [1]. The transport, uptake, and mobilization of sugars in plants are regulated by various sugar transporters, classified into monosaccharide transporters (MSTs) and sucrose transporters (SUTs) [2]. In addition, MSTs are divided into seven subfamilies. These are sugar transport proteins (STP), polyol transporters (PLT), early responsive to dehydration 6 (ERD6), inositol transporters (INT), tonoplast monosaccharide transporters (TMT), plastidic glucose transporters (pGlcT), vacuolar glucose transporters (VGT). STPs facilitate growth, development, yield, and stress tolerance in plants, regulating sugar transport [3]. STPs are involved in storing sugars in chloroplasts and vacuoles to ensure balanced cellular homeostasis [4]. PLTs are involved in the loading and unloading of polyols in the phloem to support proper plant growth under supra-optimal sugar concentrations and stress tolerance [5]. ERD6s are involved in mobilizing vacuolar glucose to maintain proper growth and stress tolerance in plants [5]. Inositol transporters, are crucial for liberating inositol from vacuoles to contribute abiotic stress tolerance [6]. TMTs and VGTs have been well characterized in *Arabidopsis*, that play vital roles in vacuolar sugar transport [7]. In maize, pGlcTs are involved in nocturnal glucose transport and have been extensively studied in several crops [8–10].

SUCs are vital membrane proteins which are responsible for loading sucrose into the phloem (source-to-sink transport), unloading it into sink (roots, fruits, seeds), and facilitating energy distribution, signaling, and stress response by moving sugars across membranes [9]. Indeed, STs have already been identified and characterized in *Arabidopsis*, passion fruit, rice, potato, tomato, grapevine, pear, apple, woodland strawberry, Chinese jujube, and longan. Therefore, molecular characterization of

maize sugar transporters is crucial for understanding their role in sugar transport and stress responsiveness, which can aid in improving plant productivity and abiotic stress tolerance.

Maize (*Zea mays* L.) is a crucial cereal crop that plays a vital role in global food security. The sugar content in maize kernels is the results from active sugar transport during the grain-filling stage and remobilization during senescence, which is essential for proper growth and stress adaptation [11]. Although sugar transporters have already been identified in maize, critical structural characterization to explore their molecular roles of sugar transport and abiotic stress tolerance have not been reported yet. Except SWEET transporters, critical characterization of other sugar transporters in maize have not been published yet. Therefore, the current research was undertaken to do potential characterization of maize sugar transporters following integration of phylogenetic analysis, structural function, synteny, molecular docking, molecular dynamics (MD) simulation, expression profiling, and experimental expression validation through qRT-PCR and subcellular localization. Indeed, MD simulations were done to validate the structural stability of the docked complex to guide the future research of biological functions of the maize sugar transporters in transporting sugars. Indeed, qRT-PCR mediated relative mRNA expression of potential sugars transporters under different tissues and salt stress were done to explore the molecular mechanism of the transporters in transporting sugars into different tissues and salt stress tolerance. The integrated dry and wet lab research might guide the future research in exploring molecular mechanism of how sugar transporters play roles in sugar transport, accumulation and mobilization along with abiotic stress tolerance.

## Materials and methods

### Identification of sugar transporters and evaluation of their physicochemical characteristics, and subcellular localization

PFAM v37.0, CDD v3.21, and SMART v10 databases were used to identify the sugar transporter proteins of maize based on the Hidden Markov Model (HMM) of the *Arabidopsis* conserved domains of sugar transporter proteins (accessed on February 15, 2025) [12]. The most recent entire maize genome (B73_RefGen_v4) sequence was extracted from the Phytozome v14 web server (https://phytozomenext.jgi.doe.gov/). The 62 known *Arabidopsis* protein sequences of sugar transporters were obtained from the TAIR10 database and utilized as a query sequence to search against the maize whole protein sequences using the local BLASTP tool in TBtools v2.363. where the e-value is set at 1e-5 and the threshold of 50% identity of similarity. Additionally, ClustalX v2.1 was used to align all of the *Arabidopsis* sugar transporter protein sequences for multiple sequence alignment. Then the HMMER web server was used to search the maize sugar transporter protein by using the alignment file. To compare and parse the findings of BLASTP and HMMER search engines for getting the exact transporter protein [13]. The PFAM, CDD, NCBI, and SMART databases were used to verify the possible members of maize sugar transporter proteins by looking for the presence of conserved domains (PF00083, PF07690, and PF13347) of maize sugar transporters to eliminate any redundant sequences. To reconfirm the existence of the conserved domains, the sequences were submitted to the Inter-Pro-Scan web server and validated the results. The ProtParam (http://web.expasy.org/protparam/) and ProtComp server (http://linux1.softberry.com/) were used to analyze the physicochemical parameters and subcellular localization of maize sugar transporter proteins, respectively (accessed on February 15, 2025) [14].

### Phylogenetic tree analysis of maize, rice, and *Arabidopsis* sugar transporters

To analyze the phylogenetic relationship between maize, rice, and *Arabidopsis*, the protein sequences of each species sugar transporters were aligned with ClustalX v2.1. The sugar transporter proteins of the aforementioned species were then used to create the phylogenetic tree setting the maximum likelihood (ML) option and a 1000 bootstrap value using the MEGAX11 program [15]. The online-based web server ITOL v7 (https://itol.embl.de/login.cgi) and Adobe Illustrator were used to generate an interactive phylogenetic tree (accessed on February 17, 2025) [16].

## Prediction of three-dimensional (3D) modeling and transmembrane domains

Using the Phyre2 v2.2 website (https://www.sbg.bio.ic.ac.uk/phyre2/html/page.cgi?id=index), the 3D structure of maize sugar transporter proteins was created in order to evaluate the structural variations and their impact on the functions of these protein families (accessed on February 20, 2025) [17]. To evaluate the transmembrane domains and helices of the maize sugar transporter proteins, topology structures were created using the HMMTOP v2.0 web server (https://hmmtop.pbrg.hu/) (accessed on January 20, 2025) [18].

## Synteny analysis of the proteins

Genome sequences of *S. bicolor, O. sativa, B. distachyon, H. vulgare*, and *A. thaliana* were downloaded from the phytozome v14 server to analyze the syntenic relationships of maize sugar transporters with the above-mentioned five genomes. In TBtools v2.363, the MCScanX was utilized to predict orthologous pairs between maize and the above-mentioned genomes, where the graphics view was utilized to visualize the output [18].

## Gene ontology (GO) analysis

To analyze the potential roles of maize sugar transporters, the corresponding IDs of candidate genes were submitted to the ShinyGO v0.77 (https://bioinformatics.sdstate.edu/go77/) databases to predict the biological and molecular functions and cellular components of maize sugar transporters (accessed on February 25, 2025). GO enrichment was computed using a p-value cut-off (FDR) of 0.05 [19].

## Docking of maize sugar transporter proteins and protein-protein interaction (PPI) network analysis

The 3D structures of the Glu ($C_6H_{12}O_6$; PubChem ID: 5793), Fru ($C_6H_{12}O_6$; PubChem ID: 2723872), galactose (Gal) ($C_6H_{12}O_6$; PubChem ID: 439357), and Suc ($C_{12}H_{22}O_{11}$; PubChem ID:5988) ligands were retrieved from the PubChem database (accessed on March 3, 2025) [20]. For the preparation of receptor proteins and ligands as well as docking analysis, Autodock 4.2 and Autodock Vina software were used [21]. Subsequent interaction analysis between maize sugar transporters and sugars was performed using PyMOL and LigPlot+ v.2.2.4 software [22].

The STRING 11.0 (https://string-db.org/) web server was used to predict the maize sugar transporter protein-protein interaction network (accessed on March 10, 2025). The protein sequences of identified maize sugar transporters were input into the STRING database, and *Zea mays* was selected as the genome [23]. The STRING tools parameters were configured as follows: network type-full STRING network: the significance of network edge evidence; the lower confidence parameter (0.150) was set as the minimum required interaction score. Subsequently, Cytoscape software was used to visualize the PPI networks [24].

## Molecular dynamics simulation

To assess the stability of the protein-ligand interaction in the chosen candidate compounds, a 100 ns molecular dynamics simulation was carried out for the complex structures using the "Desmond v6.3 Program" in Schrodinger 2020−3 under the Linux framework [25]. The TIP3P water model was employed in the process to simulate the sucrose transporters and ligand intricate. To neutralize the entire system with a salt concentration of 0.15 M, $Na^+$ and $Cl^-$ were introduced to an orthorhombic box shape that was 10 Å from the center, and the OPLS3e force field was also applied. Using the NPT ensemble at a constant pressure of 101,325 Pascals and a temperature of 300 K, the MD simulation of the protein-ligand complex system was further reduced [26]. To evaluate the stability and dynamic characteristics of the complexes, RMSD, RMSF, rGyr, SASA, and protein-ligand contact values were analyzed.

## Expression patterns at different growth stages and tissues, different abiotic stresses

To explore how the maize sugar transporters are expressed in various tissues, including the leaf, internodes, root, seed, embryo, endosperm, and reproductive tissues (tassel, silk, and cob), we obtained the RNA-seq data from the MaizeMine

v15 server (https://maizemine-v15.rnet.missouri.edu/maizemine/begin.do) (accessed on March 22, 2025) [27]. The whole genome transcripts datasets of abiotic stresses were gathered from NCBI web server for drought (GEO accession number GSE189392), heat (GEO accession number GSE122866), salinity (GEO accession number: GSE128432), and nitrogen starvation (GEO accession number GSE111425) in order to explore the stress-responsive expression patterns of these transporters (accessed on March 30, 2025) [14]. Expression patterns at different growth stages and tissues, and different abiotic stresses. After retrieving the expression values of sugar transporters from the above-mentioned datasets, log2FC values were calculated for a uniform representation. Differential expression analysis was considered significant when |log2FC| ≥ 1 and adjusted p-value (FDR) ≤ 0.05. RNA-seq datasets contained at least three biological replicates per condition. The statistical processing and visualization were conducted using SPSS and TBtools v2.363, respectively [28].

## Building co-expression network of maize sugar transporters

The ATTED-II v.11.1 (https://atted.jp/) web server was used to conduct the co-expression analysis of the maize sugar transporter, selecting a few genes at the PPI option and the many option for the coex (accessed on April 2, 2025). After entering the sugar transporters Entrez IDs on the server, the analysis was carried out against the genome of maize [14]. Gene co-expression relationships were determined using default parameters of the server, correlation coefficient ≥ 0.70 and p-value ≤ 0.05 were retained for network construction.

## Relative mRNA expression analysis through quantitative real-time PCR (qRT-PCR)

Maize inbred line B73 was grown in Chongzhou, Sichuan Province, China, under natural soil conditions to examine tissue-specific expression patterns of *ZmST* genes. In addition, B73 seedlings were cultivated in controlled growth chambers under a 12-hour light (30 °C) and 12-hour dark (22 °C) photoperiod, with a light intensity of approximately 500 µmol m$^{-2}$ s$^{-1}$. Tissue samples tassel, husk, silk, leaf, internode, node, and root were collected 70 days after sowing, whereas kernel samples were harvested two weeks following pollination. Potential genes (*ZmPLT1, 8, ZmSTP1, ZmTMT1,* and *ZmSUC3*) were then examined for expression patterns in those tissues using qRT-PCR analysis. To analyze the salinity-induced expression of the above-mentioned genes, seedling roots were drowned in a solution of 50 mM NaCl [29].

TRIpure Reagent (Aidlab Biotechnologies TiangenBiotech Co., Ltd., Beijing, China) was used to extract maize (B73) RNAs. One microgram of RNA was reverse-transcribed using HiScript® III All-in-one RT SuperMix reverse transcription kit reagents (Vazyme Biotech Co., Ltd.). One mL of Enzyme Mix, 4 µl of 5×All-in-one RT SuperMix, 1 µg of RNA, and a suitable quantity of RNase-free ddH$_2$O were utilized to create cDNA and remove gDNA. The mixture was incubated at 50° C for 15 minutes, and then it was raised to 85° C for 5 seconds. For qPCR, the total reaction volume was 10 µL. The qPCR Master Mix (2×) contains 5 µL, 1.0 µL of cDNA, 0.5 µL of forward primer, 0.5 µL of reverse primer, and 3.0 µL of ddH$_2$O. ZmElf1 served as an internal reference in the qRT-PCR experiment, which was carried out using the primers listed in S1 Table. Here, qPCR was carried out using the CFX96TM real-time equipment (Bio-Rad, Hercules, CA, USA) and the ChamQ Universal SYBR qPCR Master Mix (Vazyme Biotech Co., Ltd., Nanjing, China) [30]. Each qRT-PCR experiment was conducted using three biological replicates and three technical replicates per sample. Expression levels were presented as mean ± standard deviation (SD). Statistical significance between treatments and controls was determined using Student's t-test. Differences were considered statistically significant at $p < 0.05$. The expression of CT values was analyzed using the $2^{-\Delta\Delta CT}$ technique [31].

## Subcellular localization

Two recombinant construct pCAMBIA2300-ZmPLT1-GFP and pCAMBIA2300-ZmPLT8-GFP, were generated by inserting ZmPLT1 and ZmPLT8 CDSs without stop codons at the upstream of GFP protein in pCAMBIA2300-GFP vector via Sma I sites. Marker vector pCAMBIA2300-35S-Histone-mCherry-NOS was used for nuclear localization. The above-mentioned

two recombinant vectors, along with a negative control of empty vector (pCAMBIA2300::GFP), were injected into *Agrobacterium* strain GV3101 and transiently transformed the resuspension solution into tobacco leaves. A laser confocal scanning microscope (Olympus, Japan) was used to observe the distribution of GFP fluorescence [32]. Primers used for cloning and expression of genes are mentioned in S1 Table. Subcellular localization experiments were repeated independently at least three times to ensure consistency of fluorescence signal patterns.

## Statistical analysis

All statistical analyses were conducted using GraphPad Prism v9.0 and TBtools v2.363. Results are presented as mean±SD. Statistical differences were evaluated using Student's t-test and one-way ANOVA, and significance was set at $p < 0.05$.

## Results

### Identification of sugar transporters and evaluation of their physicochemical characteristics, and subcellular localization

In the maize genome, 60 sugar transporters were identified and validated following a series of critical bioinformatics analyses utilizing the whole genome sequence. Among these, 16 polyol transporters (PLT), 15 sugar transporter proteins (STP), 9 early-response to dehydration six-like (ERD6L), 8 inositol transporters (INT), 5 sucrose carriers (SUC), 3 tonoplast monosaccharide transporters (TMT), 2 plastidic glucose transporters (pGlcT), and 2 vacuolar glucose transporter (VGT) proteins were identified in maize (Table 1). Significant variations in physicochemical characteristics among these proteins might reveal their functional diversity and structural integrity (Table 1). Among the transporter proteins, significant variation of amino acid length (the smallest ZmERD6L-9 having 71 amino acids and longest ZmTMT1 having 746 amino acids) might reveal the significant structural alteration that might alter the protein folding and functional properties of the respective proteins. Additionally, significant variation in the molecular weight of these proteins (8.1 kD in ZmERD6L-9 to 80.95 kD in ZmTMT2) might reveal their functional diversity in the maize genome. Due to having isoelectric points (pI) values of >5, these proteins might be basic in nature. Additionally, the positive GRAVY scores among all 60 proteins (0.192 in ZmSUC2 to 0.812 inZmERD6L-8), might guide the hydrophobic characteristics of all the proteins. Except ZmERD6L-7, ZmINT2, ZmINT4, and ZmpGlcT2, plasma membrane localization of 56 sugar transporters might reveal their major roles in transporting intercellular sugar (Table 1).

### Phylogenetic tree analysis of maize, rice, and *Arabidopsis* sugar transporters

To explore the evolutionary origin of maize sugar transporter from its ancestors, we have utilized the 60 protein sequences of maize, 69 rice, and 62 *Arabidopsis* sugar transporters and constructed and constructed a ML system phylogenetic tree (Fig 1). The topology of the phylogenetic tree classified the 191 full-length sugar transporter proteins into eight groups, VGT, pGlcT, TMT, INT, SUC, PLT ERD6 and STP. Among these eight, highest members (57) in theSTP subfamily across the three species, might represent their conserved, diverse and essential roles in transporting sugars. Notably, maize and rice contain a higher number of STP (15) members compared to *Arabidopsis*, suggesting their lineage-specific expansion in monocots, possibly due to increased demands for sugar allocation in complex vegetative and reproductive tissues (Fig 1). Similarly, the higher enrichment of PLT (16) and INT (8) subfamilies in maize might reflect gene duplication during maize evolution that may have diverse functional alteration (Fig 1). In contrast, the follower members in SUC, TMT, pGlcT, and VGT subfamilies suggest a precise functional conservation with limited expansion (Fig 1). Overall, the differential members among sugar transporter gene families in maize, rice, and *Arabidopsis* highlights evolutionary divergence shaped by species-specific physiological requirements and adaptation strategies.

**Table 1. Characterization of identified sugar transporters in maize.**

| Protein IDs | Protein Name | Uniprot IDs | Amino acid number | Protein Molecular Weight(kDa) | pI | GRAVY | Subcellular Localization |
|---|---|---|---|---|---|---|---|
| Zm00001d021935_P001 | ZmPLT1 | A0A1D6IHX6 | 514 | 54.83 | 9.36 | 0.597 | Plasma membrane |
| Zm00001d029645_P001 | ZmPLT2 | A0A1D6K6F3 | 520 | 55.29 | 9.53 | 0.558 | Plasma membrane |
| Zm00001d028144_P001 | ZmPLT3 | B4FQN6 | 526 | 56.03 | 7.71 | 0.353 | Plasma membrane |
| Zm00001d021936_P002 | ZmPLT4 | A0A1D6IHX8 | 513 | 54.03 | 8.53 | 0.662 | Plasma membrane |
| Zm00001d021942_P001 | ZmPLT5 | A0A1D6IHY2 | 495 | 52.5 | 8.88 | 0.645 | Plasma membrane |
| Zm00001d006688_P001 | ZmPLT6 | A0A1D6EZM7 | 517 | 54.36 | 9.16 | 0.621 | Plasma membrane |
| Zm00001d006697_P001 | ZmPLT7 | A0A1D6EZN4 | 521 | 55.07 | 9.16 | 0.589 | Plasma membrane |
| Zm00001d021938_P001 | ZmPLT8 | B4FF54 | 514 | 54.25 | 9 | 0.612 | Plasma membrane |
| Zm00001d048178_P001 | ZmPLT9 | A0A1D6PIC1 | 532 | 56.73 | 8.85 | 0.422 | Plasma membrane |
| Zm00001d028151_P001 | ZmPLT10 | A0A1D6JSE9 | 525 | 56.15 | 9.08 | 0.444 | Plasma membrane |
| Zm00001d002864_P001 | ZmPLT11 | B6TAG3 | 534 | 57.52 | 6.21 | 0.455 | Plasma membrane |
| Zm00001d001818_P001 | ZmPLT12 | A0A1D6DT50 | 550 | 57.36 | 6.67 | 0.541 | Plasma membrane |
| Zm00001d001817_P001 | ZmPLT13 | A0A1D6DT49 | 539 | 55.88 | 6.68 | 0.667 | Plasma membrane |
| Zm00001d048776_P001 | ZmPLT14 | K7UPD5 | 488 | 50.93 | 8.89 | 0.658 | Plasma membrane |
| Zm00001d023939_P001 | ZmPLT15 | A0A1D6IWW8 | 479 | 50.16 | 8.75 | 0.742 | Plasma membrane |
| Zm00001d048771_P001 | ZmPLT16 | B4FDN6 | 482 | 50.2 | 8.79 | 0.73 | Plasma membrane |
| Zm00001d028230_P001 | ZmSTP1 | A0A1D6JTD3 | 515 | 56.68 | 9.05 | 0.507 | Plasma membrane |
| Zm00001d020463_P002 | ZmSTP2 | A0A1D6I499 | 524 | 56.2 | 9.34 | 0.557 | Plasma membrane |
| Zm00001d044245_P001 | ZmSTP3 | A0A1D6NJW4 | 510 | 55.64 | 9.54 | 0.672 | Plasma membrane |
| Zm00001d032409_P001 | ZmSTP4 | B8A122 | 537 | 56.84 | 9.35 | 0.601 | Plasma membrane |
| Zm00001d005594_P001 | ZmSTP5 | A0A1D6EP34 | 519 | 56.63 | 9.16 | 0.527 | Plasma membrane |
| Zm00001d019138_P001 | ZmSTP6 | A0A1D6HVS2 | 514 | 56.13 | 9.04 | 0.483 | Plasma membrane |
| Zm00001d020071_P001 | ZmSTP7 | A0A1D6I1X6 | 523 | 57.26 | 9.41 | 0.47 | Plasma membrane |
| Zm00001d007078_P001 | ZmSTP8 | A0A1D6F305 | 491 | 53.55 | 9.36 | 0.654 | Plasma membrane |
| Zm00001d027268_P001 | ZmSTP9 | B4FS29 | 526 | 57.49 | 9.21 | 0.515 | Plasma membrane |

*(Continued)*

**Table 1.** (Continued)

| Protein IDs | Protein Name | Uniprot IDs | Amino acid number | Protein Molecular Weight(kDa) | pl | GRAVY | Subcellular Localization |
|---|---|---|---|---|---|---|---|
| Zm00001d045395_P001 | ZmSTP10 | A0A1Q1CLX2 | 519 | 54.51 | 9.82 | 0.581 | Plasma membrane |
| Zm00001d053846_P001 | ZmSTP11 | A0A1D6QSS3 | 552 | 59.92 | 8.92 | 0.648 | Plasma membrane |
| Zm00001d025572_P001 | ZmSTP12 | K7UIW9 | 508 | 53.52 | 9.39 | 0.661 | Plasma membrane |
| Zm00001d032906_P001 | ZmSTP13 | B4FSQ6 | 509 | 54.31 | 9.14 | 0.554 | Plasma membrane |
| Zm00001d003471_P001 | ZmSTP14 | A0A1D6E9C5 | 558 | 59.53 | 9.85 | 0.522 | Plasma membrane |
| Zm00001d025573_P001 | ZmSTP15 | A0A1D6J7U0 | 457 | 49.86 | 9.71 | 0.627 | Plasma membrane |
| Zm00001d008374_P001 | ZmERD6L-1 | A0A1D6FCA5 | 490 | 51.87 | 5.67 | 0.644 | Plasma membrane |
| Zm00001d040243_P002 | ZmERD6L-2 | A0A1D6MPE6 | 493 | 52.35 | 8.68 | 0.616 | Plasma membrane |
| Zm00001d039051_P002 | ZmERD6L-3 | A0A1D6MD64 | 510 | 54.69 | 8.74 | 0.526 | Plasma membrane |
| Zm00001d009669_P003 | ZmERD6L-4 | C0P753 | 507 | 54.08 | 9.23 | 0.635 | Plasma membrane |
| Zm00001d009605_P002 | ZmERD6L-5 | A0A1D6FKM9 | 500 | 52.91 | 9.24 | 0.613 | Plasma membrane |
| Zm00001d039052_P001 | ZmERD6L-6 | A0A1D6MD70 | 411 | 44.02 | 9.17 | 0.543 | Plasma membrane |
| Zm00001d051186_P001 | ZmERD6L-7 | A0A1D6Q5H6 | 246 | 26.87 | 7.66 | 0.36 | NONE |
| Zm00001d008867_P001 | ZmERD6L-8 | A0A1D6FGF9 | 82 | 9.17 | 4.52 | 0.812 | Plasma membrane |
| Zm00001d045901_P001 | ZmERD6L-9 | A0A1D6NZS4 | 71 | 8.1 | 4.31 | 0.731 | Plasma membrane |
| Zm00001d018803_P001 | ZmINT1 | A0A1D6HSI3 | 586 | 62.44 | 8.82 | 0.361 | Plasma membrane |
| Zm00001d025834_P001 | ZmINT2 | A0A1D6JAA1 | 582 | 62.8 | 8.62 | 0.392 | Vacuole |
| Zm00001d019537_P001 | ZmINT3 | A0A1D6HYA1 | 486 | 51.91 | 9.06 | 0.454 | Plasma membrane |
| Zm00001d025749_P002 | ZmINT4 | A0A1D6J968 | 534 | 56.68 | 5.87 | 0.583 | Vacuole |
| Zm00001d015897_P001 | ZmINT5 | A0A1D6H4J3 | 236 | 25.75 | 9.73 | 0.524 | Plasma membrane |
| Zm00001d035374_P001 | ZmINT6 | A0A1D6LG74 | 154 | 16.53 | 10.48 | 0.743 | Plasma membrane |
| Zm00001d014051_P001 | ZmINT7 | A0A1D6GPN6 | 154 | 16.5 | 10.28 | 0.779 | Plasma membrane |
| Zm00001d034214_P001 | ZmINT8 | A0A1D6L677 | 154 | 16.59 | 10.53 | 0.755 | Plasma membrane |
| Zm00001d041192_P003 | ZmSUC1 | A0A1D6MV11 | 502 | 53.37 | 8.84 | 0.486 | Plasma membrane |
| Zm00001d018527_P002 | ZmSUC2 | A0A1D6HQ38 | 631 | 68.17 | 8.57 | 0.192 | Plasma membrane |
| Zm00001d027854_P002 | ZmSUC3 | C4J5U5 | 521 | 55.14 | 8.67 | 0.602 | Plasma membrane |

*(Continued)*

**Table 1.** (Continued)

| Protein IDs | Protein Name | Uniprot IDs | Amino acid number | Protein Molecular Weight(kDa) | pI | GRAVY | Subcellular Localization |
|---|---|---|---|---|---|---|---|
| Zm00001d048311_P003 | ZmSUC4 | A0A1D6PJ66 | 529 | 56.11 | 8.57 | 0.599 | Plasma membrane |
| Zm00001d033011_P001 | ZmSUC5 | B4FX10 | 509 | 53.52 | 7.46 | 0.584 | Plasma membrane |
| Zm00001d016274_P011 | ZmTMT1 | A0A096PY37 | 746 | 79.82 | 5.26 | 0.397 | Plasma membrane |
| Zm00001d014872_P001 | ZmTMT2 | C0PE06 | 764 | 80.95 | 4.72 | 0.316 | Plasma membrane |
| Zm00001d048823_P007 | ZmTMT3 | A0A1D6PQ88 | 653 | 71.7 | 5.64 | 0.354 | Plasma membrane |
| Zm00001d014435_P004 | ZmVGT1 | A0A1D6GTC8 | 543 | 58.41 | 6.16 | 0.601 | Plasma membrane |
| Zm00001d012938_P002 | ZmVGT2 | A0A1D6GE24 | 622 | 65.9 | 10.28 | 0.466 | Plasma membrane |
| Zm00001d039973_P003 | ZmpGlcT1 | A0A1D6MM04 | 540 | 56.8 | 9.22 | 0.567 | Plasma membrane |
| Zm00001d020374_P003 | ZmpGlcT2 | A0A1D6I3U3 | 618 | 65.71 | 7.97 | 0.49 | NONE |

## Three-dimensional modeling and transmembrane topology analysis

Following the structural prediction of maize sugar transporter proteins, we have analyzed their 3D configurations to identify the arrangement of polypeptide chains. All proteins have multiple alpha helices, transmembrane helices, and coil topologies, that are essential for sugar transport across membranes. The top proteins with the highest confidence (100%) and coverage in 3D structure, highlight their key structural features that support their functional integrity in transporting sugars (Fig 2A). Significant variations in alpha helix (59% in ZmTMT3 and ZmpGlcT1to 69% inZmSTP15 and ZmSUC4) might represent the structural stability and diverse active biding sites (Table 2). Following that, significant variations in transmembrane helix among eight representative sugar transporter proteins of all sub families that have best coverage and confidence during 3D structural modeling (ZmTMT3 and ZmVGT1 exhibited 48%, ZmERD6L-2, ZmINT4, and ZmpGlcT1 49%, ZmPLT16 and ZmSTP15 54%, and ZmSUC4 58%) might represent the diverse structural integrity and functional diversity (Table 2). These differential structural variations and functional diversity of maize sugar transporters might reveal their diverse and precise function in transporting sugar molecules in maize.

Following 3D protein characterization, multiple transmembrane helices in transmembrane topology, where existence of 12 helices in ZmERDL-2, ZmINT4, ZmpGlcT1, ZmPLT16, ZmSUC4, and ZmVGT1 proteins might again confirm their functional variation that could influence transport efficiency and substrate specificity. Indeed, longer intracellular sequences at the N- and C- terminal (Fig 2B), suggesting roles in protein interactions and localization. The existence of longer intracellular (cytoplasmic) sequences at the N- and C-termini in transmembrane protein topology might guide their crucial functional roles in providing accessible epitopes or tags for detection and orientation determination. Accordingly, cytoplasmic termini in most proteins along with extracellular and cytoplasmic termini in ZmSTP15 and ZmTMT3 might guide the possible dual roles in transport and signaling of sugars by the transporters. Significant variation in protein length among the best eight representative protein structures ranged from (456 inZmSTP15) to 652 inZmTMT3) reveal their functional diversity and substrate specificity (Table 3).

## Syneny analyses

To explore the evolutionary origin, synteny analysis of maize sugar transporters were done against *S. bicolor, B. distachyon, O. sativa, H. vulgare*, and *A. thaliana* genomes. The highest homolog pairs of maize sugar transporters with *S. bicolor*, following *O. sativa, B. distachyon, H. vulgare*, and *A. thaliana* genomes might reveal their duplication events with the genomes

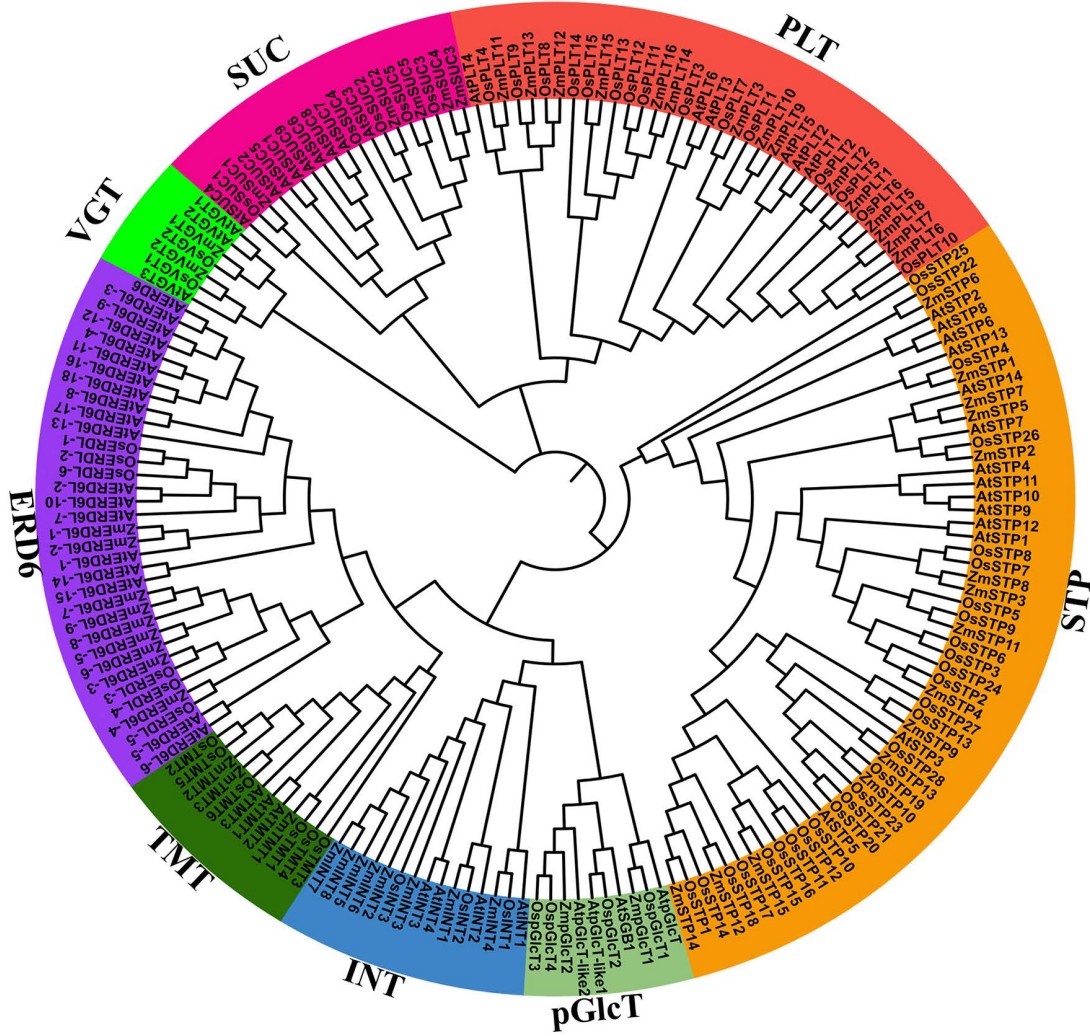

**Fig 1. Phylogenetic analysis of maize sugar transporters.** The phylogenetic tree topology was generated through MEGA11 with the maximum likelihood (ML) method and 1000 bootstrap replications. Branches corresponding to partitions reproduced in <50% bootstrap replicates are collapsed. The evolutionary distance matrices were computed using the JTT matrix-based method. **Note: PLT:** Polyol Transporter, **STP:** Sugar Transport protein, **ERD6L:** Early Response to Dehydration 6 Like, **INT:** Inositol Transporter, **SUC:** Sucrose Carrier, **TMT:** Tonoplastic Monosaccharide Transporter, **VGT:** Vacuolar Glucose Transporter, **pGlcT:** Plastidic Glucose Translocator.

(Fig 3). A notable number of 39, 29, 29, 20, and 2 syntenic pairs of maize sugar transporter with *S. bicolor B. distachyon, O. sativa, H. vulgare,* and *A. thaliana* respectively might reveal their gene duplications with the genomes (Fig 3, S2 Table). The higher number of syntenic gene pairs of maize with other monocots suggests the conserved genomic orientation during evolution of sugar transporters in monocots, whereas reduced synteny with *A. thaliana* indicates lineage-specific rearrangements or gene loss following divergence. Overall, these patterns support partial conservation of sugar transporters among different plant genomes accompanied by evolutionary diversity across different plant genomes.

## Functional annotation of maize sugar transporters through GO analysis

To explore the diverse biological functions of maize sugar transporters, gene ontology analysis was done. Significant involvement of sugar transporters in various biological processes, molecular functions, and cellular components might

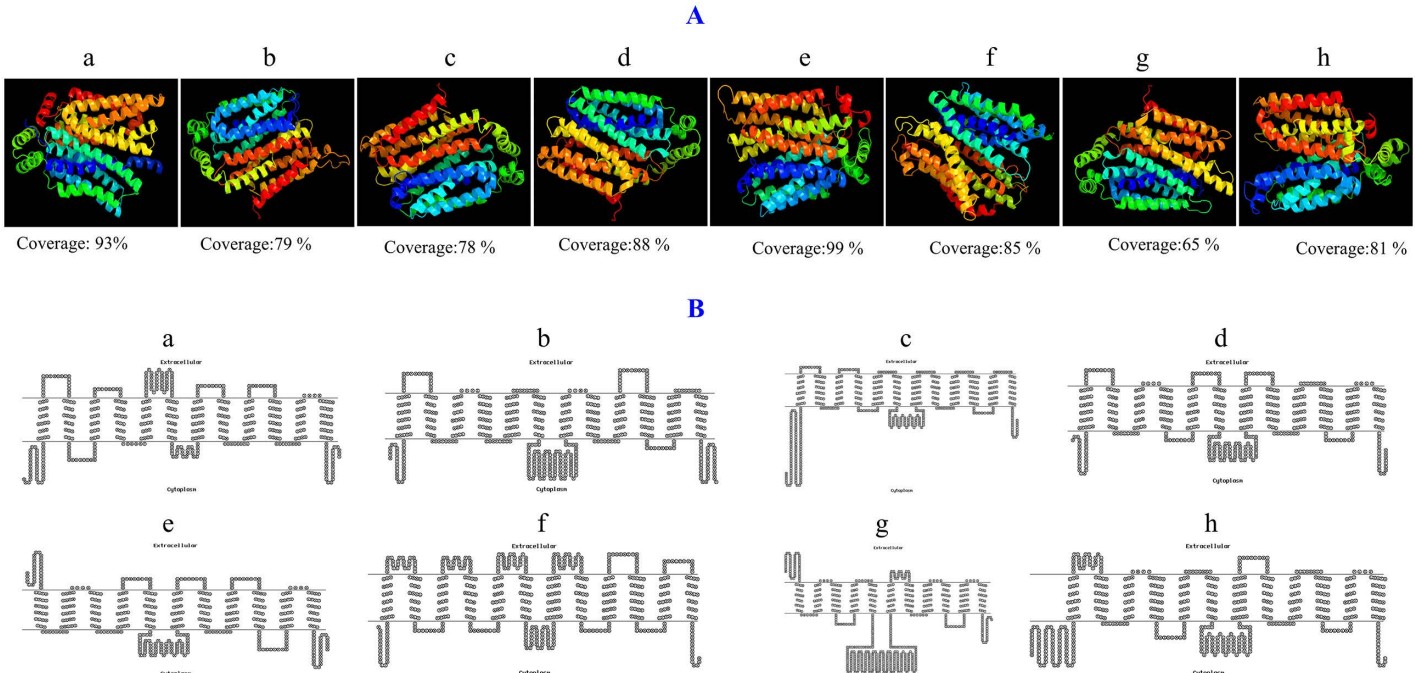

**Fig 2. Three-dimensional (3D) protein structure (A) and transmembrane topology (B) of selected eight maize sugar transporter proteins.** Each protein showed a 100% confidence score in 3D modeling. Here, a=ZmERD6L-2 (Early Response to Dehydration6 L-2), b=ZmINT4 (InositolTransporter4), c=ZmpGlcT1 (Plastidic Glucose Transporter1), d=ZmPLT6 (Polyol Transporter6), e=ZmSTP15 (Sugar Transporter Protein15), f=ZmSUC4 (Sucrose Carrier4), g=ZmTMT3 (Tonoplastic Monosaccharide Transporter3), h=ZmVGT1 (Vacuolar Glucose Transporter1).

reveal their potential involvement in regulating different metabolic process along with transporting sugars. Significant regulation of different biological process interacting sucrose, polyol, inositol, disaccharide, and oligosaccharide transport in maize might reveal their sugar mediated regulatory role in different biological process (Fig 4A, S3 Table). The regulation of golgi-network, vacuole membrane, and plasma membrane transport at molecular level might guide their deep regulatory roles in the mentioned systems (Fig 4B, S3 Table), following cellular trafficking. At cellular level, regulation of inositol transmembrane transporter activity, carbohydrate proton, and cation transmembrane transporter activity by the mentioned sugar transporters (Fig 4C, S3 Table), might guide their essential function in ion and sugar transport across the membranes.

## The molecular docking and protein-protein interaction (PPI) of maize sugar transporters

Protein-ligand docking is a molecular modeling technique used to predict how a protein and ligands interact, fit and position within the proteins binding pocket to assess binding affinity. The molecular docking of maize sugar transporters with three monosaccharides (Gul, Fru, Gal) and one disaccharide (Suc) was performed to determine the exact binding site of the sugar ligands with the maize sugar transporters. In molecular docking the lowest mean binding energy ($\Delta$G) of Suc (−5.82 kcal/mol) when complexed with 60 sugar transporters might reveal its strong binding affinity that is transported by the analyzed sugar transporters. Among these 60 transporters, ZmSTP12 and ZmSTP14 showed the lowest $\Delta$G with Fru (−6.9 kcal/mol) and ZmVGT1 with Suc (−7.1 kcal/mol), reflecting their high specificity for these sugars (Table 4, Fig 5A-D). ZmSTP14 showed lower hydrogen bonding but higher docking integrity which may suggests its stability in complexing and transporting Fru for its efficient transport.

Hydrogen bonds are critical for non-covalent interactions in protein-ligand docking that stabilize complexes, dictate binding specificity, and influence binding affinity. In our findings, hydrogen bond numbers ranged from 3 to 11 for Glu, suggesting strong,

**Table 2. Characterization of three-dimensional modeling of maize sugar transporters.**

| Protein Name | Alpha helix (%) | Beta strand (%) | TM helix (%) | Disordered (%) | Confidence (%) | Coverage (%) | Template |
|---|---|---|---|---|---|---|---|
| ZmPLT1 | 66 | 0 | 51 | 12 | 100 | 82 | c4ldsB |
| ZmPLT2 | 63 | 0 | 51 | 12 | | 81 | c4ldsB |
| ZmPLT3 | 64 | 0 | 50 | 13 | | 80 | c4ldsB |
| ZmPLT4 | 63 | 0 | 51 | 11 | | 82 | c4ldsB |
| ZmPLT5 | 69 | 0 | 53 | 10 | | 85 | c4ldsB |
| ZmPLT6 | 64 | 0 | 51 | 12 | | 82 | c4ldsB |
| ZmPLT7 | 63 | 0 | 51 | 13 | | 81 | c4ldsB |
| ZmPLT8 | 65 | 0 | 51 | 12 | | 82 | c4ldsB |
| ZmPLT9 | 63 | 0 | 50 | 12 | | 79 | c4ldsB |
| ZmPLT10 | 64 | 0 | 50 | 12 | | 80 | c4ldsB |
| ZmPLT11 | 64 | 0 | 49 | 17 | | 87 | c6h7dA |
| ZmPLT12 | 61 | 0 | 47 | 19 | | 77 | c4ldsB |
| ZmPLT13 | 63 | 0 | 50 | 18 | | 78 | c4ldsB |
| ZmPLT14 | 67 | 0 | 54 | 10 | | 86 | c4ldsB |
| ZmPLT15 | 70 | 0 | 55 | 10 | | 88 | c4ldsB |
| ZmPLT16 | 67 | 0 | 54 | 9 | | 88 | c4ldsB |
| ZmSTP1 | 64 | 1 | 52 | 12 | | 94 | c6h7dA |
| ZmSTP2 | 62 | 0 | 51 | 13 | | 93 | c6h7dA |
| ZmSTP3 | 65 | 0 | 52 | 10 | | 95 | c6h7dA |
| ZmSTP4 | 62 | 0 | 50 | 14 | | 91 | c6h7dA |
| ZmSTP5 | 66 | 0 | 52 | 12 | | 93 | c6h7dA |
| ZmSTP6 | 64 | 0 | 52 | 11 | | 93 | c6h7dA |
| ZmSTP7 | 64 | 0 | 52 | 12 | | 93 | c6h7dA |
| ZmSTP8 | 66 | 0 | 52 | 11 | | 95 | c6h7dA |
| ZmSTP9 | 63 | 0 | 52 | 12 | | 92 | c6h7dA |
| ZmSTP10 | 65 | 0 | 52 | 11 | | 93 | c6h7dA |
| ZmSTP11 | 66 | 0 | 48 | 12 | | 88 | c6h7dA |
| ZmSTP12 | 66 | 0 | 54 | 9 | | 95 | c6h7dA |
| ZmSTP13 | 67 | 0 | 54 | 8 | | 95 | c6h7dA |
| ZmSTP14 | 61 | 0 | 48 | 13 | | 87 | c6h7dA |
| ZmSTP15 | 69 | 0 | 54 | 5 | | 99 | c6h7dA |
| ZmERD6L-1 | 68 | 0 | 55 | 14 | | 89 | c7wsmA |
| ZmERD6L-2 | 65 | 0 | 49 | 10 | | 93 | c4ybqB |
| ZmERD6L-3 | 64 | 0 | 52 | 15 | | 85 | c7wsmA |
| ZmERD6L-4 | 66 | 0 | 52 | 16 | | 86 | c7wsmA |
| ZmERD6L-5 | 65 | 0 | 53 | 15 | | 87 | c7wsmA |
| ZmERD6L-6 | 69 | 0 | 55 | 12 | | 88 | c7wsmA |
| ZmERD6L-7 | 58 | 1 | 52 | 8 | 99.5 | 78 | c4ldsB |
| ZmERD6L-8 | 79 | 0 | 44 | 0 | 95.8 | 99 | c4pypA |
| ZmERD6L-9 | 70 | 0 | 31 | 1 | 91.5 | 85 | c4pypA |
| ZmINT1 | 59 | 2 | 57 | 10 | 100 | 72 | c4ldsB |
| ZmINT2 | 66 | 0 | 57 | 11 | | 72 | c4ldsB |
| ZmINT3 | 67 | 0 | 62 | 6 | | 78 | c7wsmA |
| ZmINT4 | 64 | 0 | 49 | 12 | | 79 | c4ldsB |
| ZmINT5 | 66 | 0 | 54 | 16 | 99.8 | 81 | c4ldsB |

*(Continued)*

Table 2. (Continued)

| Protein Name | Alpha helix (%) | Beta strand (%) | TM helix (%) | Disordered (%) | Confidence (%) | Coverage (%) | Template |
|---|---|---|---|---|---|---|---|
| ZmINT6 | 76 | 0 | 54 | 4 | 99.9 | 99 | c4ldsB |
| ZmINT7 | 76 | 0 | 54 | 4 | | 99 | c4ldsB |
| ZmINT8 | 77 | 0 | 56 | 3 | | 99 | c4ldsB |
| ZmSUC1 | 70 | 1 | 61 | 10 | 100 | 94 | c8bb6A |
| ZmSUC2 | 56 | 0 | 49 | 14 | | 68 | c8bb6A |
| ZmSUC3 | 70 | 0 | 58 | 9 | | 92 | c8bb6A |
| ZmSUC4 | 69 | 0 | 58 | 9 | | 85 | c4ikyA |
| ZmSUC5 | 71 | 0 | 61 | 10 | | 93 | c8bb6A |
| ZmTMT1 | 54 | 2 | 44 | 12 | | 58 | c4gbzA |
| ZmTMT2 | 49 | 2 | 42 | 18 | | 55 | c4ldsB |
| ZmTMT3 | 59 | 0 | 48 | 9 | | 65 | c4ldsB |
| ZmpGlcT1 | 59 | 3 | 49 | 18 | | 78 | c4ldsB |
| ZmpGlcT2 | 57 | 0 | 42 | 17 | | 68 | c4ldsB |
| ZmVGT1 | 62 | 0 | 48 | 18 | | 81 | c5c65A |
| ZmVGT2 | 53 | 0 | 42 | 24 | | 67 | c4ldsB |

**Table 3. Characterization of transmembrane topology of representative of eight subfamilies of maize sugar transporters.**

| Transmembrane helices | ZmPLT16 | ZmSTP15 | ZmERD6L-2 | ZmINT4 | ZmSUC4 | ZmTMT3 | ZmVGT1 | ZmpGclT1 |
|---|---|---|---|---|---|---|---|---|
| TM1 | 28–52 | 31–53 | 40–59 | 31–55 | 26–44 | 47–67 | 83–102 | 104–123 |
| TM2 | 69–87 | 62–86 | 80–99 | 74–93 | 71–90 | 74–93 | 133–155 | 136–160 |
| TM3 | 96–114 | 91–109 | 118–135 | 102–121 | 103–121 | 98–122 | 164–181 | 169–187 |
| TM4 | 119–141 | 118–137 | 148–167 | 126–149 | 148–167 | 139–158 | 186–210 | 198–215 |
| TM5 | 154–173 | 152–171 | 174–193 | 158–182 | 180–199 | 165–184 | 227–244 | 226–250 |
| TM6 | 188–207 | 238–257 | 239–258 | 191–210 | 230–249 | 435–454 | 253–271 | 259–277 |
| TM7 | 278–300 | 272–292 | 285–307 | 311–330 | 294–318 | 481–503 | 345–369 | 341–365 |
| TM8 | 315–337 | 301–322 | 322–341 | 335–358 | 349–368 | 510–534 | 386–404 | 374–393 |
| TM9 | 346–368 | 337–357 | 350–369 | 367–386 | 381–403 | 539–558 | 413–432 | 402–424 |
| TM10 | 377–400 | 378–399 | 384–407 | 407–431 | 422–446 | 577–601 | 441–465 | 433–457 |
| TM11 | 413–436 | 404–425 | 416–434 | 444–468 | 459–478 | 606–625 | 478–497 | 470–492 |
| TM12 | 441–460 | | 439–463 | 477–496 | 493–512 | | 502–526 | 501–518 |
| Length of amino acids | 481 | 456 | 492 | 533 | 528 | 652 | 542 | 539 |

stable binding, for efficient transport of glucose (Table 4). ZmSTP6 bound with Glu tilizing11 hydrogen bonds might suggest its specialized role in Glu transport, which is stabilized by significant hydrophobic interactions, and crucial for membrane integration (Fig 5A). Similarly, for Fru, ZmSTP14 complexed using 10 hydrogen bonds, that might optimize its interaction for efficient Fru transporting (Table 4; Fig 5B). For complex formation of sugar transporters with Gal, hydrogen bonds varied from 2 to 11, that might reflect the diverse roles in Gal transport (Table 4). The 10 hydrogen bonds ZmTMT3-Gal docked complex might suggest the specificity of the protein for Gal, which is supported by hydrophobic interactions, crucial for transporter stability (Fig 5C). The 3–13 hydrogen bonds ranged for Suc when complexed with 60 sugar transporters might from, indicate differential affinities of the transporters in transporting Suc (Table 4). Indeed, the highest 13 hydrogen bonds in ZmSUC1-Suc docked complex might suggest strong binding affinity of the transporter in transporting Suc which is tabilized by hydrophobic interactions, that enhance

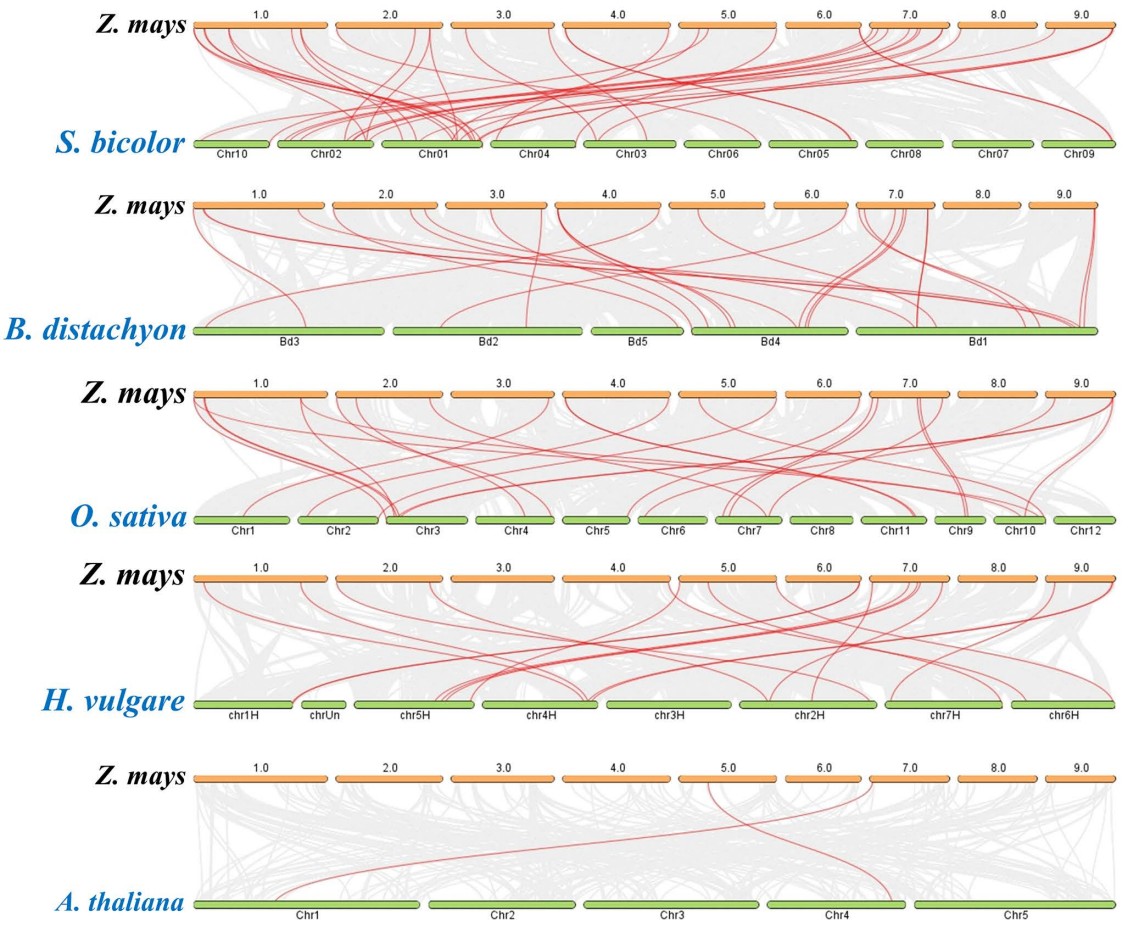

**Fig 3. Syntenic relationships of maize sugar transporters.** The syntenic relationships were calculated with Sorghum, Brachypodium, Rice, Barley, and Arabidopsis genomes. The gray lines in the background represent the collinear blocks within maize sugar transporter and other plant genomes, while the red lines highlight the syntenic gene pairs. **Note: PLT:** Polyol Transporter, **STP:** Sugar Transport protein, **ERD6L:** Early Response to Dehydration 6 Like, **INT:** Inositol Transporter, **SUC:** Sucrose Carrier, **TMT:** Tonoplastic Monosaccharide Transporter, **VGT:** Vacuolar Glucose Transporter, **pGlcT:** Plastidic Glucose Translocator.

membrane stability (Fig 5D). Interestingly, Glu residues were common in hydrogen bonds across all docked complexes, which indicate their regulatory roles in stabilizing the transporters (Fig 5A-D).

Protein-protein interaction plays a key role in predicting the protein function of target protein with others protein. In our research, the PPI analysis identified 60 proteins (nodes) and 1007 interactions (edges), having a local clustering coefficient of 0.741 and an enrichment p-value of <1.0e-16. This suggests a dense, biologically significant network with strong clustering, pointing to potential functional relationships between the proteins. The PPI enrichment p-value of <1.0e-16 shows that the results are highly significant and interactions are biologically meaningful. Among all, 59 transporters showed significant interaction (Fig 5E), highlighting that maize sugar transporters are greatly involved in sugar transport and mobilization.

## Molecular dynamics simulation

Molecular dynamics simulation analyses were performed for 100 ns to evaluate the structural stability and dynamic behavior of four best protein–ligand complexes having highest binding affinity, namely ZmSTP14–Fructose,

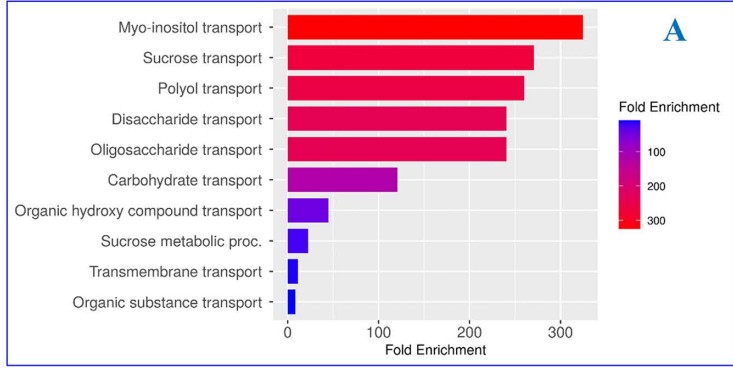
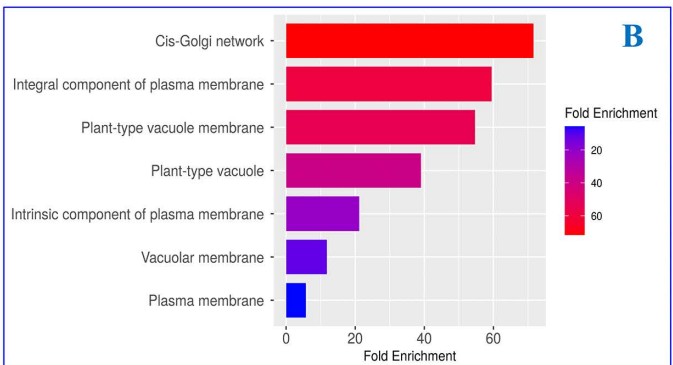
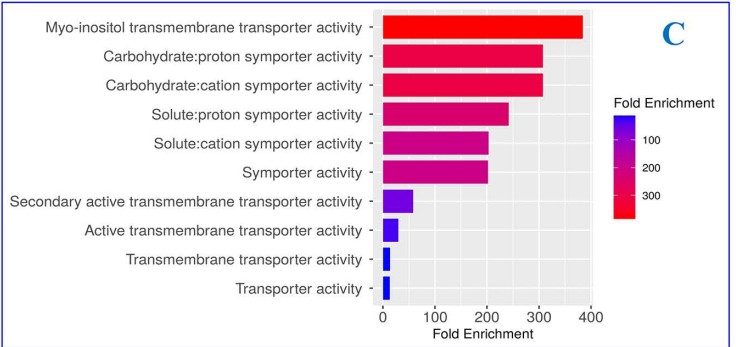

**Fig 4. Gene ontology analysis of sugar transporters in maize.** Transporters that influence biological processes (A) molecular functions (B) and cellular components (C) are presented here.

ZmSTP10–Galactose, ZmSTP10–Glucose, and ZmVGT1–Sucrose. Structural deviations were assessed using root mean square deviation (RMSD) analysis of Cα atoms (Fig 6A). ZmSTP14–Fructose, ZmSTP10–Galactose, and ZmSTP10–Glucose complexes exhibited RMSD values of 3.16 Å, 3.36 Å, and 2.83 Å, respectively, indicating the complexes are highly stable. In contrast, the ZmVGT1–Sucrose complex showed higher fluctuations with an RMSD of 5.11 Å, suggesting less binding stability.

Residue-level flexibility was further evaluated using root mean square fluctuation (RMSF) analysis (Fig 6B). Higher fluctuations at the N- and C-termini were observed, that might indicate the protein flexibility typical for functional dynamics. The average RMSF values of 1.54 Å (ZmSTP14–Fructose), 1.21 Å (ZmSTP10–Galactose), 1.05 Å (ZmSTP10–Glucose), and 2.20 Å (ZmVGT1–Sucrose), might suggest the stable ligand binding with localized flexibility. The highest peak fluctuations were observed for each complex at SER188, ASP294, and GlY314 residues for ZmSTP14-Fructose, following ILE261, ILE484, and PRO507 residues for ZmSTP10-Galactose, ILE196, ALA250, and GLY385 residues for ZmSTP10-Glucose, and LEU128, PHE184, TYR243, GLY287, GLN306, SER374, LEU435, THR505, and LYS531 residues for ZmVGT1-Sucrose, respectively. These identified key flexible residues might play vital roles in ligand interaction and transport function.

The compactness of the protein–ligand complexes was assessed through radius of gyration (Rg) analysis (Fig 6C). The four complexes of ZmSTP14-Fructose, ZmSTP10-Galactose, ZmSTP10-Glucose, and ZmVGT1-Sucrose showed the Rg value of 2.379 Å to 2.198 Å (average 0.181 Å), 2.439 Å to 2.267 Å (average 0.172 Å), 2.506 Å to 2.222 Å (average 0.284 Å) and 3.320 Å to 3.097 Å (average 0.223 Å), respectively. All complexes showed stable Rg values with limited fluctuations, might indicate the insignificant conformational changes in docked complexes over time, which suggest structural integrity of the complexes. Interestingly, the largest Rg range in ZmVGT1–Suc complex might reflect the larger conformational binding dynamics in the complex.

**Table 4. Docking analysis of maize sugar transporters showing the binding energies and number of hydrogen bonds formed in each transporter-sugar interaction.**

| Sl. No. | Maize sugar transporter protein | Sugar (Ligand) | | | | | | | |
|---|---|---|---|---|---|---|---|---|---|
| | | Fructose | | Galactose | | Glucose | | Sucrose | |
| | | Binding energy (ΔG (kcal/mol)) | H-bonds | Binding energy (ΔG (kcal/mol)) | H-bonds | Binding energy (ΔG (kcal/mol)) | H-bonds | Binding energy (ΔG (kcal/mol)) | H-bonds |
| 1 | ZmPLT1 | −5.30 | 8 | −5.00 | 5 | −5.00 | 6 | −6.0 | 8 |
| 2 | ZmPLT2 | −5.10 | 7 | −5.00 | 5 | −5.40 | 7 | −5.5 | 4 |
| 3 | ZmPLT3 | −5.00 | 4 | −5.00 | 5 | −5.40 | 5 | −5.5 | 6 |
| 4 | ZmPLT4 | −5.00 | 3 | −5.40 | 7 | −5.80 | 7 | −4.9 | 8 |
| 5 | ZmPLT5 | −5.00 | 4 | −5.50 | 8 | −5.80 | 9 | −5.7 | 7 |
| 6 | ZmPLT6 | −5.10 | 4 | −5.20 | 5 | −5.50 | 6 | −5.8 | 8 |
| 7 | ZmPLT7 | −5.30 | 5 | −5.10 | 9 | −5.50 | 5 | −5.4 | 5 |
| 8 | ZmPLT8 | −4.90 | 3 | −5.20 | 5 | −5.50 | 6 | −5.3 | 7 |
| 9 | ZmPLT9 | −5.50 | 5 | −5.90 | 8 | −5.50 | 5 | −6.0 | 10 |
| 10 | ZmPLT10 | −5.30 | 5 | −5.70 | 4 | −5.60 | 8 | −6.9 | 11 |
| 11 | ZmPLT11 | −6.00 | 7 | −6.10 | 8 | −6.10 | 9 | −6.6 | 9 |
| 12 | ZmPLT12 | −5.70 | 8 | −5.50 | 7 | −5.40 | 7 | −5.8 | 9 |
| 13 | ZmPLT13 | −5.00 | 9 | −4.90 | 6 | −5.30 | 6 | −5.9 | 8 |
| 14 | ZmPLT14 | −5.90 | 6 | −5.10 | 7 | −5.40 | 5 | −5.6 | 5 |
| 15 | ZmPLT15 | −5.30 | 6 | −5.70 | 6 | −5.80 | 8 | −6.1 | 5 |
| 16 | ZmPLT16 | −5.20 | 7 | −5.20 | 8 | −5.50 | 10 | −5.4 | 6 |
| 17 | ZmSTP1 | −5.90 | 6 | −5.30 | 3 | −5.80 | 8 | −5.3 | 10 |
| 18 | ZmSTP2 | −5.20 | 7 | −5.30 | 6 | −5.60 | 8 | −6.3 | 4 |
| 19 | ZmSTP3 | −5.00 | 5 | −5.20 | 8 | −6.30 | 11 | −6.3 | 10 |
| 20 | ZmSTP4 | −5.50 | 9 | −5.30 | 5 | −5.50 | 5 | −6.0 | 10 |
| 21 | ZmSTP5 | −5.30 | 4 | −5.30 | 7 | −5.30 | 4 | −5.6 | 8 |
| 22 | ZmSTP6 | −5.90 | 5 | −5.90 | 6 | −6.30 | 11 | −5.1 | 6 |
| 23 | ZmSTP7 | −5.30 | 4 | −5.30 | 7 | −5.60 | 6 | −5.8 | 8 |
| 24 | ZmSTP8 | −6.20 | 9 | −5.70 | 8 | −6.10 | 10 | −6.7 | 7 |
| 25 | ZmSTP9 | −6.00 | 7 | −4.90 | 2 | −5.40 | 6 | −5.5 | 12 |
| 26 | ZmSTP10 | −6.60 | 9 | −6.60 | 10 | −6.80 | 10 | −6.9 | 8 |
| 27 | ZmSTP11 | −5.70 | 8 | −5.30 | 8 | −5.70 | 7 | −6.7 | 7 |
| 28 | ZmSTP12 | −6.90 | 11 | −6.50 | 9 | −6.50 | 8 | −5.8 | 12 |
| 29 | ZmSTP13 | −6.10 | 9 | −6.40 | 9 | −6.50 | 10 | −6.5 | 8 |
| 30 | ZmSTP14 | −6.90 | 10 | −6.40 | 7 | −5.20 | 6 | −5.4 | 6 |
| 31 | ZmSTP15 | −5.40 | 5 | −6.20 | 9 | −5.70 | 5 | −6.8 | 7 |
| 32 | ZmERD6L-1 | −5.50 | 6 | −6.10 | 10 | −5.60 | 6 | −6.2 | 5 |
| 33 | ZmERD6L-2 | −5.80 | 5 | −5.80 | 6 | −5.70 | 6 | −5.3 | 5 |
| 34 | ZmERD6L-3 | −6.10 | 8 | −5.60 | 7 | −5.80 | 5 | −5.8 | 8 |
| 35 | ZmERD6L-4 | −5.70 | 5 | −5.50 | 4 | −5.70 | 7 | −6.5 | 7 |
| 36 | ZmERD6L-5 | −5.70 | 7 | −5.40 | 8 | −5.60 | 8 | −6.4 | 7 |
| 37 | ZmERD6L-6 | −5.70 | 9 | −5.60 | 7 | −6.20 | 9 | −6.4 | 8 |
| 38 | ZmERD6L-7 | −5.60 | 6 | −5.10 | 4 | −5.40 | 6 | −5.2 | 4 |
| 39 | ZmERD6L-8 | −4.00 | 8 | −3.90 | 7 | −3.80 | 5 | −4.2 | 6 |
| 40 | ZmERD6L-9 | −3.50 | 7 | −3.50 | 7 | −3.50 | 9 | −3.9 | 5 |
| 41 | ZmINT1 | −5.80 | 9 | −5.50 | 6 | −5.70 | 6 | −5.4 | 5 |
| 42 | ZmINT2 | −6.00 | 9 | −6.00 | 8 | −6.10 | 8 | −6.1 | 6 |

*(Continued)*

**Table 4.** (Continued)

| Sl. No. | Maize sugar transporter protein | Sugar (Ligand) | | | | | | | |
|---|---|---|---|---|---|---|---|---|---|
| | | Fructose | | Galactose | | Glucose | | Sucrose | |
| | | Binding energy (ΔG (kcal/mol)) | H-bonds | Binding energy (ΔG (kcal/mol)) | H-bonds | Binding energy (ΔG (kcal/mol)) | H-bonds | Binding energy (ΔG (kcal/mol)) | H-bonds |
| 43 | ZmINT3 | −5.40 | 7 | −5.50 | 8 | −5.60 | 10 | −5.9 | 9 |
| 44 | ZmINT4 | −5.40 | 8 | −5.60 | 8 | −5.40 | 8 | −6.9 | 5 |
| 45 | ZmINT5 | −4.50 | 8 | −5.30 | 3 | −5.40 | 5 | −4.8 | 4 |
| 46 | ZmINT6 | −5.10 | 6 | −5.00 | 6 | −5.10 | 6 | −5.1 | 6 |
| 47 | ZmINT7 | −5.10 | 6 | −5.20 | 7 | −5.10 | 7 | −5.3 | 6 |
| 48 | ZmINT8 | −4.70 | 7 | −4.30 | 3 | −4.30 | 5 | −5.0 | 5 |
| 49 | ZmSUC1 | −5.20 | 6 | −5.50 | 6 | −5.80 | 5 | −5.9 | 13 |
| 50 | ZmSUC2 | −5.00 | 5 | −5.20 | 8 | −5.30 | 8 | −5.9 | 7 |
| 51 | ZmSUC3 | −5.10 | 8 | −5.40 | 3 | −5.40 | 6 | −6.1 | 7 |
| 52 | ZmSUC4 | −5.10 | 6 | −4.70 | 6 | −5.60 | 6 | −5.6 | 7 |
| 53 | ZmSUC5 | −5.20 | 3 | −5.40 | 4 | −5.40 | 3 | −6.0 | 5 |
| 54 | ZmTMT1 | −5.90 | 8 | −6.00 | 8 | −4.90 | 7 | −5.4 | 3 |
| 55 | ZmTMT2 | −5.30 | 7 | −5.30 | 9 | −5.40 | 4 | −6.6 | 9 |
| 56 | ZmTMT3 | −5.10 | 9 | −6.30 | 11 | −5.30 | 6 | −5.9 | 9 |
| 57 | ZmVGT1 | −5.70 | 6 | −5.40 | 7 | −5.60 | 7 | −7.1 | 6 |
| 58 | ZmVGT2 | −5.80 | 6 | −5.60 | 3 | −5.90 | 4 | −6.1 | 10 |
| 59 | ZmpGlcT1 | −5.30 | 7 | −6.50 | 9 | −5.50 | 5 | −6.1 | 10 |
| 60 | ZmpGlcT2 | −5.40 | 6 | −5.20 | 3 | −5.00 | 9 | −5.1 | 8 |

**Note:** ΔG, binding energy (kcal/mol); H-bonds, number of hydrogen bonds between maize sugar transporter proteins and corresponding sugar ligands.

Accordingly, the solvent-accessible surface area (SASA) analysis was conducted to examine the changes in hydrophobic and hydrophilic surface exposure upon ligand binding (Fig 6D). The SASA value of the ZmSTP14-Fructose, ZmSTP10-Galactose, ZmSTP10-Glucose, and ZmVGT1-Sucrose complexes ranged from 0.472 to 336.507, 18.742 to 343.493, 17.210 to 346.857, and 19.287 to 108.043 Å2, respectively, indicate variation of degrees during hydrophobic and hydrophilic surface exposures. Indeed, the respective complexes showed the average SASA value fluctuations of 120.08, 196.69, 284.77, and 58.23 Å2, respectively., These variations might suggest that the biomolecules have higher folds of amino acids exposed to the surface area that is accessible to molecular solvents. Interestingly, the larger fluctuations in ZmSTP10-Glu complexes might suggest their undergoing greater surface area exposure during ligand binding, that could potentially influence their transport function.

## Expression patterns of maize sugar transporters under different growth stages and tissues, and different abiotic stresses

In the B73 maize inbred line, significant variation in expression of sugar transporters was observed across different tissues. Among PLTs, significant higher expression of *ZmPLT4, ZmPLT8, ZmPLT10, ZmPLT11,* and *ZmPLT15* in the internode, leaf, and root, indicate their crucial roles in sugar loading and unloading, which is essential for energy distribution in these tissues. Conversely, significant lower expression of *ZmPLT12, ZmPLT13,* and *ZmPLT14* in the above conditions might suggest their more specialized or less prominent roles in sugar transport. *ZmSTP1, ZmSTP2,* and *ZmSTP10* were highly expressed in the leaf and root, which may highlight their importance in sugar transport during source to sink. Notably, *ZmERD6L-4* showed moderate expression in all tissues, suggesting their involvement in transporting sugar routinely

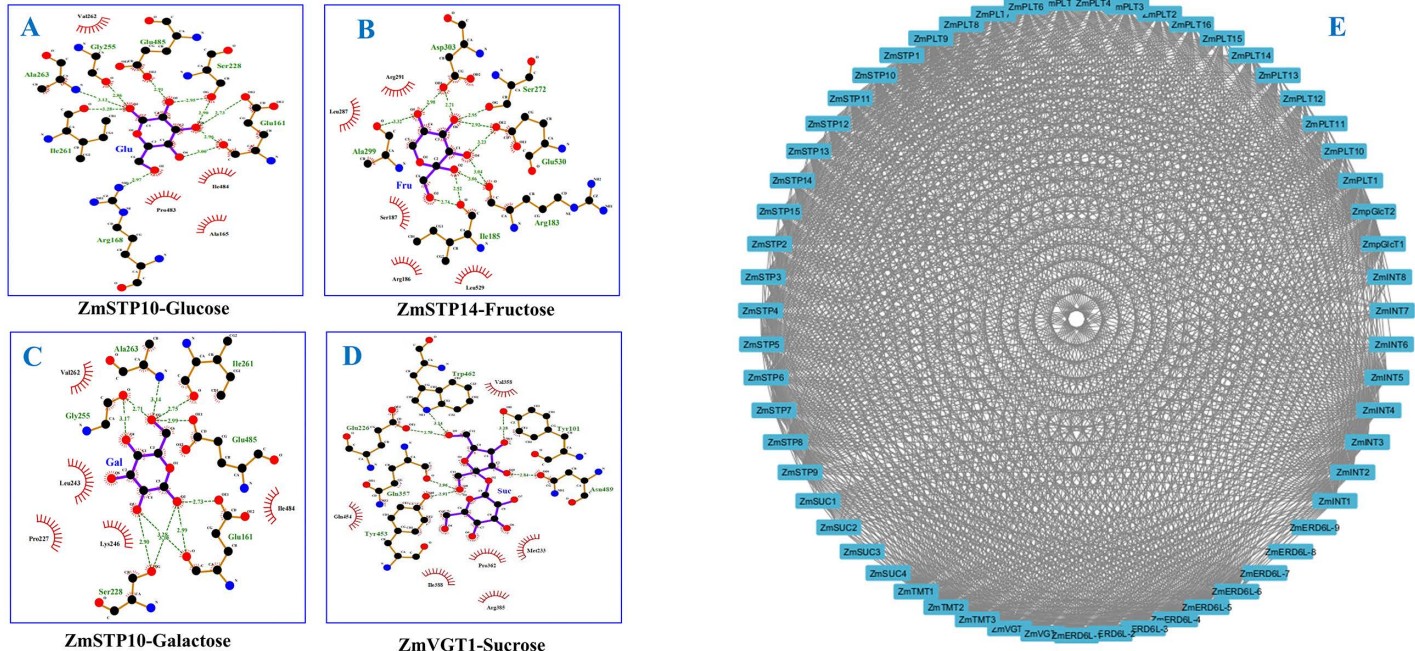

**Fig 5. The molecular docking (A-D) and protein-protein interaction (E) of maize sugar transporters.** The three monosaccharide ligands glucose, fructose, and galactose docked with ZmSTP10 **(A)**, ZmSTP14 **(B)**, ZmSTP10 **(C)**, and disaccharide sucrose was docked with ZmVGT1 **(D)**. **Note: PLT:** Polyol Transporter, **STP:** Sugar Transport protein, **ERD6L:** Early Response to Dehydration 6 Like, **INT:** Inositol Transporter, **SUC:** Sucrose Carrier, **TMT:** Tonoplastic Monosaccharide Transporter, **VGT:** Vacuolar Glucose Transporter, **pGlcT:** Plastidic Glucose Translocator..

transport or stress response across the plant. Significant higher expression of *ZmERD6L-2* in the anthers might reveal its potential roles inpollen development or reproductive processes, while *ZmERD6L-6* showed specific expression in the root system, associated with root sugar uptake and transport (Fig 7, S4 Table).

The expression of sugar transporter was significantly altered under different abiotic stresses. Under drought stress, *ZmSUC3* was most highly expressed in the leaves, highlighting its critical role in sugar transport during water deficit, which is essential for maintaining cellular energy balance. *ZmPLT8, ZmSTP1, ZmERD6L-1, ZmINT1,* and *ZmTMT3* were significantly upregulated under both cuticular and leaf under water stress environments, which could suggest their involvement in water regulation and stress response (Fig 8A, S5 Table). Under salinity stress, sugar transporters in the salinity-tolerant genotype (ST) showed higher upregulation compared to the sensitive genotype (SS), indicating a potential adaptation mechanism to maintain sugar transport and cellular function in salt tolerant genotypes. *ZmSUC3* showed upregulation in both genotypes under the stress, which could suggest its fundamental role in stress tolerance (Fig 8B, S6 Table). Nitrogen starvation led to decreased expression of most sugar transporters, whereas, *ZmSTP9, ZmERD6L-3, ZmSUC1,* and *ZmVGT1* showed higher expression under both high and low nitrogen conditions, reflecting their involvement in regulating sugar transport and metabolism to adapt to nitrogen limitation (Fig 9A, S7 Table). Similarly, under heat stress, *ZmSTP1, ZmERD6L-3, ZmSUC1,* and *ZmVGT2* were significantly upregulated in leaves, where *ZmSUC3* being highly expressed across all leaf regions, which indicate its key role in maintaining sugar supply and cellular function during heat stress (Fig 9B, S8 Table).

## Co-expression analysis of maize sugar transporters

In the co-expression analysis, 491 genes were found to be co-expressed with 43 maize sugar transporters and were grouped into four clusters (Fig 10). The KEGG ontology of these co-expressing genes with the above-mentioned

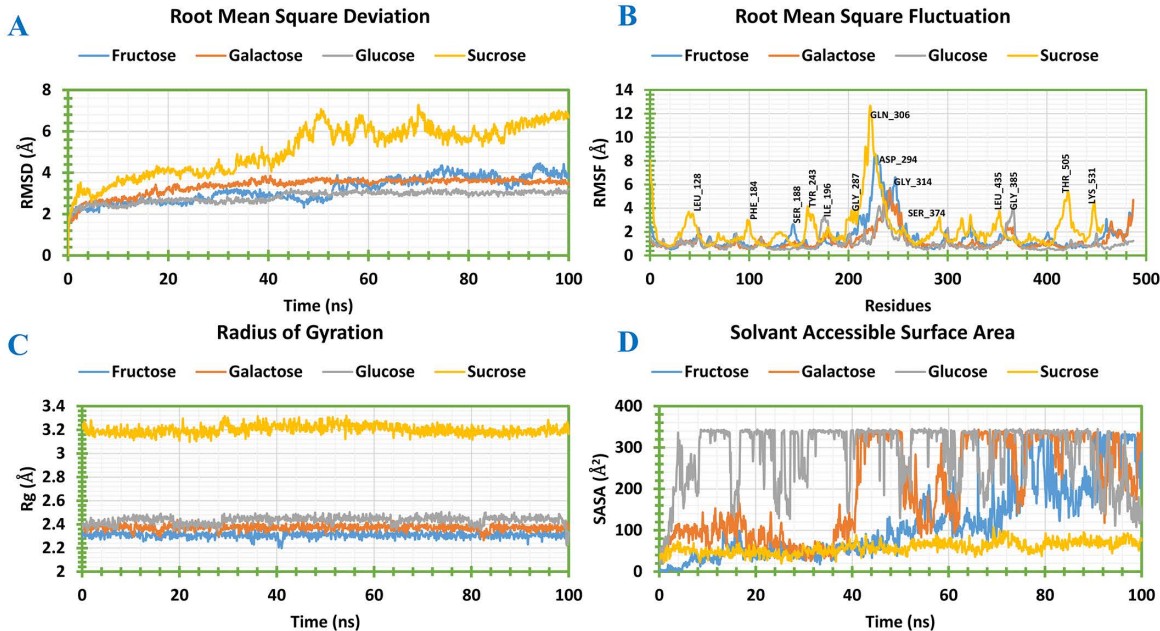

**Fig 6. Molecular dynamics simulation of the selected ligand-protein complexes deliberated from a 100 ns simulation.** (A) the extracted RMSD values **(B)** The RMSF values from Ca atoms of the protein-ligand docked complex. **(C)** The Rg values, and (D) the SASA values of the protein-ligand complexes. Here, ZmSTP14-Fructose, ZmSTP10-Galactose, ZmSTP10-Glucose, and ZmVGT1-Sucrose complexes were characterized by blue, orange, grey, and yellow color separately for the RMSD, RMSF, Rg and SASA's figure.

sugar transporters showed that they are involved in phenylpropanoid biosynthesis (KEGG: zma00940; 11 genes), starch and sucrose metabolism (KEGG: zma00500; 10 genes), plant hormone signal transduction (KEGG: zma04075; 10 genes), carbon metabolism (KEGG: zma01200; 9 genes), and biosynthesis of amino acids (KEGG: zma01230; 9 genes) (S1 Fig, S9 Table). These identified pathways and co-expressed genes suggest the integral role of sugar transporters in regulating plant growth, stress responses, and energy metabolism interacting different signaling pathway jointly.

Interestingly, the drought-regulated protein glutathione transferase19 (LOC541834) co-expressed with *ZmINT1* (LOC100501657) (Fig 10A), suggesting a role in drought tolerance through redox regulation. The nutrient transporters, nitrate transporter 2 (LOC542092) and phosphate transporter protein 2 (LOC732716), co-expressed with *ZmSTP5* (LOC100282609) and *ZmPLT1* (LOC103633356) and *ZmPLT4* (LOC100381931), indicate their involvement in coordinated sugar transport while balancing other nutrients (Fig 10A). Co-expression of maize sugar transporters with the stress-associated transcription factors, stress-responsive protein LOC100283176 with *ZmSTP14* (LOC100193044), MYB8 (LOC100125635) with *ZmSTP9* (LOC100283177), and ZAT12 (LOC103634472) with *ZmINT16* (LOC100193068), highlights sugar transporters play critical role in stress tolerance, regulating stress response pathways (Fig 10A). Co–expression of heat stress responsive transcription factors A-6b (LOC109621226) with *ZmPLT10* (LOC100274325) might link the involvement of sugar transporters in regulating heat stress tolerance (Fig 10B). In addition, co-expression of Zinc finger protein (LOC100273252) with *ZmINT2* (LOC100383930) and *ZmSTP12* (LOC100283132), might indicate the transporter mediated transcriptional regulation in stress adaptation or vice-versa (Fig 10B). Indeed, the two remaining clusters, *ZmPLT15* and *ZmERD6L-4*, showed co-expression with uncharacterized proteins, which may suggest their potential novel roles in stress tolerance or plant development (Fig 10C-D).

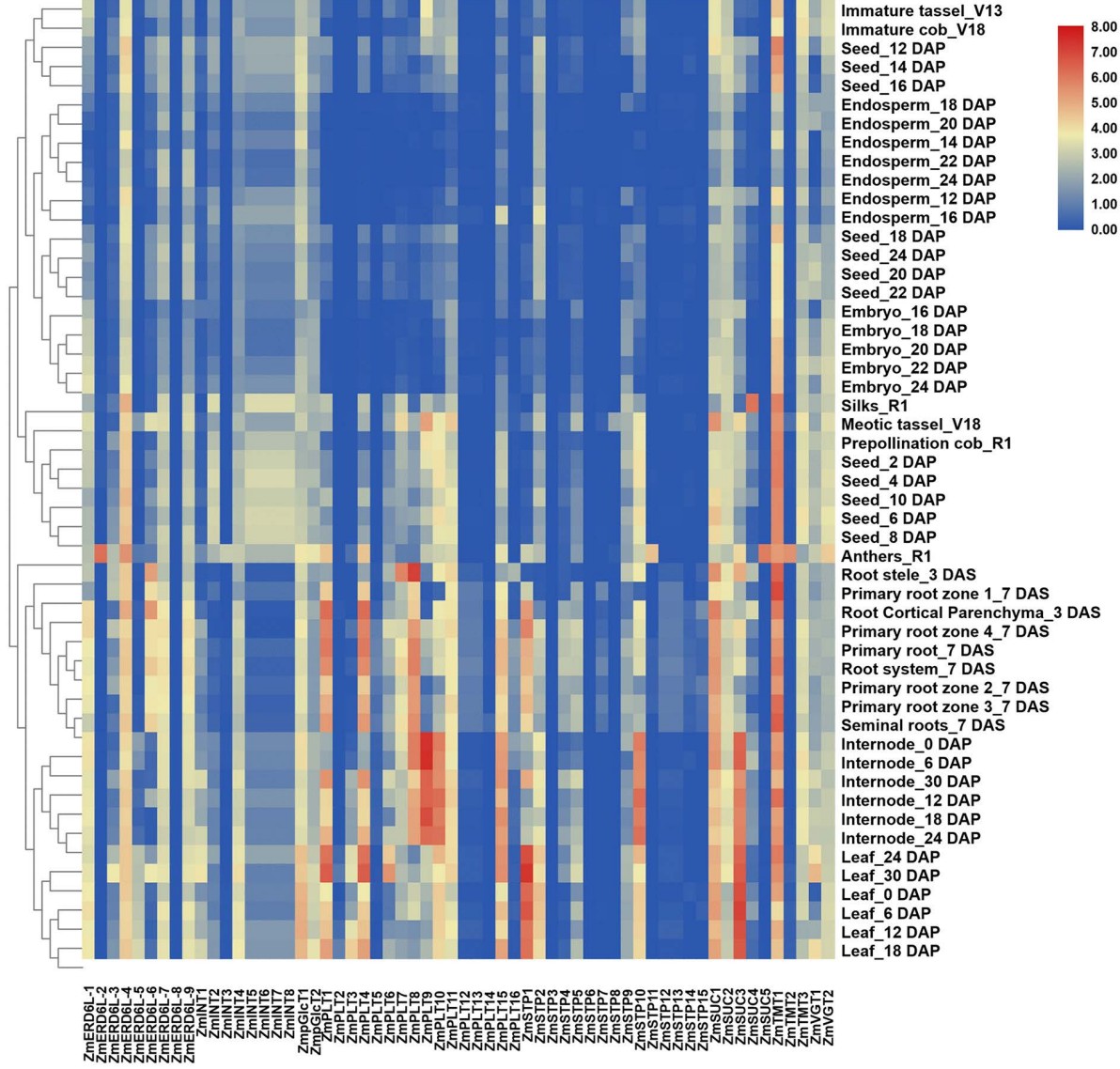

**Fig 7. Tissue-specific expression analysis of maize sugar transporters in different tissues. Note: PLT:** Polyol Transporter, **STP:** Sugar Transport protein, **ERD6L:** Early Response to Dehydration 6 Like, **INT:** Inositol Transporter, **SUC:** Sucrose Carrier, **TMT:** Tonoplastic Monosaccharide Transporter, **VGT:** Vacuolar Glucose Transporter, **pGlcT:** Plastidic Glucose Translocator.

### Relative mRNA expression analysis through quantitative real-time PCR (qRT-PCR)

Expression of the five sugar transporter genes (*ZmSTP1, ZmPLT1, 8, ZmTMT1,* and *ZmSUC3*) were cross-validated using qRT-PCR. Except for *ZmTMT1*, all four transporters were significantly upregulated at tassel, suggesting their critical role in sugar transport during reproductive development (Fig 11). The same trend of upregulation was detected in *ZmPLT8* at husk, node, and root, *ZmSTP1* at leaf and root, *ZmSUC3* at leaf and node, and *ZmPLT1* at node, which indicate their issue-specific roles in sugar loading and transport (Fig 11). Interestingly, *ZmTMT1* exhibited higher expression in all eight analyzed tissues, suggesting it may have a more universal function in sugar transport across various plant organs.

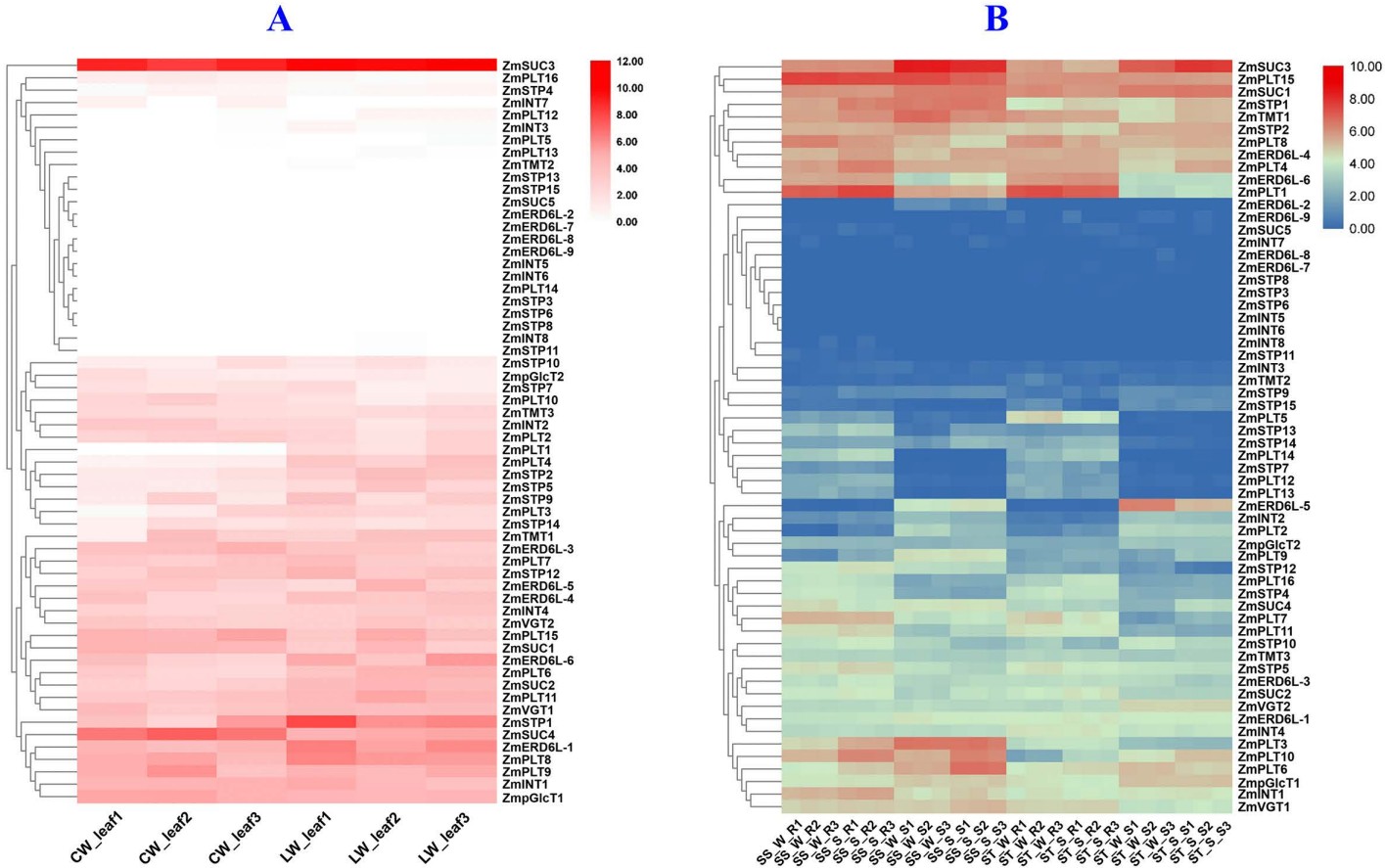

**Fig 8. Expression analysis of maize sugar transporters under drought (A) and salinity stress conditions (B). Note: PLT:** Polyol Transporter, **STP:** Sugar Transport protein, **ERD6:** Early Response to Dehydration 6, **INT:** Inositol Transporter, **SUC:** Sucrose Carrier, **TMT:** Tonoplastic Monosaccharide Transporter, **VGT:** Vacuolar Glucose Transporter, **pGlcT:** Plastidic Glucose Translocator. CW = Control water, LW = Low water, SSWR = Salinity sensitive watered root, SSSR = Salinity sensitive stressed (water) root, SSWS = Salinity sensitive watered shoot, SSSS = Salinity sensitive stressed (water) shoot, STWR = Salinity tolerant watered root, STSR = Salinity tolerant stressed (saline water) root, STWS = Salinity tolerant watered shoot, STSS = Salinity tolerant stressed (saline water) shoot. 1,2,3 represent independent replications.

To evaluate how salinity affects expression of maize sugar transporters, relative mRNA expression of the above mentioned five maize sugar transporters in root and shoot tissues was observed under salt stress. Notable upregulation of *ZmSTP1* (7-fold increase), *ZmTMT1* (0.9-fold increase), and *ZmPLT1* (0.4-fold increase) was observed in root tissues, suggesting their role in maintaining sugar transport and osmotic balance under salinity stress. Conversely, *ZmSUC3* expression was downregulated in the root (0.5-fold decrease) and unchanged in the shoot, indicate reduced sugar transport in response to stress. Interestingly, *ZmPLT8* (0.9-fold decrease), *ZmSTP1* (0.3-fold decrease), and *ZmTMT1* (0.5-fold decrease) were downregulated in shoot tissues, which may potentially reflect a shift in resource allocation to the roots under stress (Fig 12). The altered expression of these genes might help maize in adapting salinity stress regulating osmotic balance and modulating stress-related signaling pathways.

## Subcellular localization

We have predicted the subcellular localization of maize sugar transporters based on a relevant web server (Table 1). According to the prediction, except ZmERD6L-7, ZmINT2, ZmINT4, and ZmpGlcT2, all were plasma membrane localized.

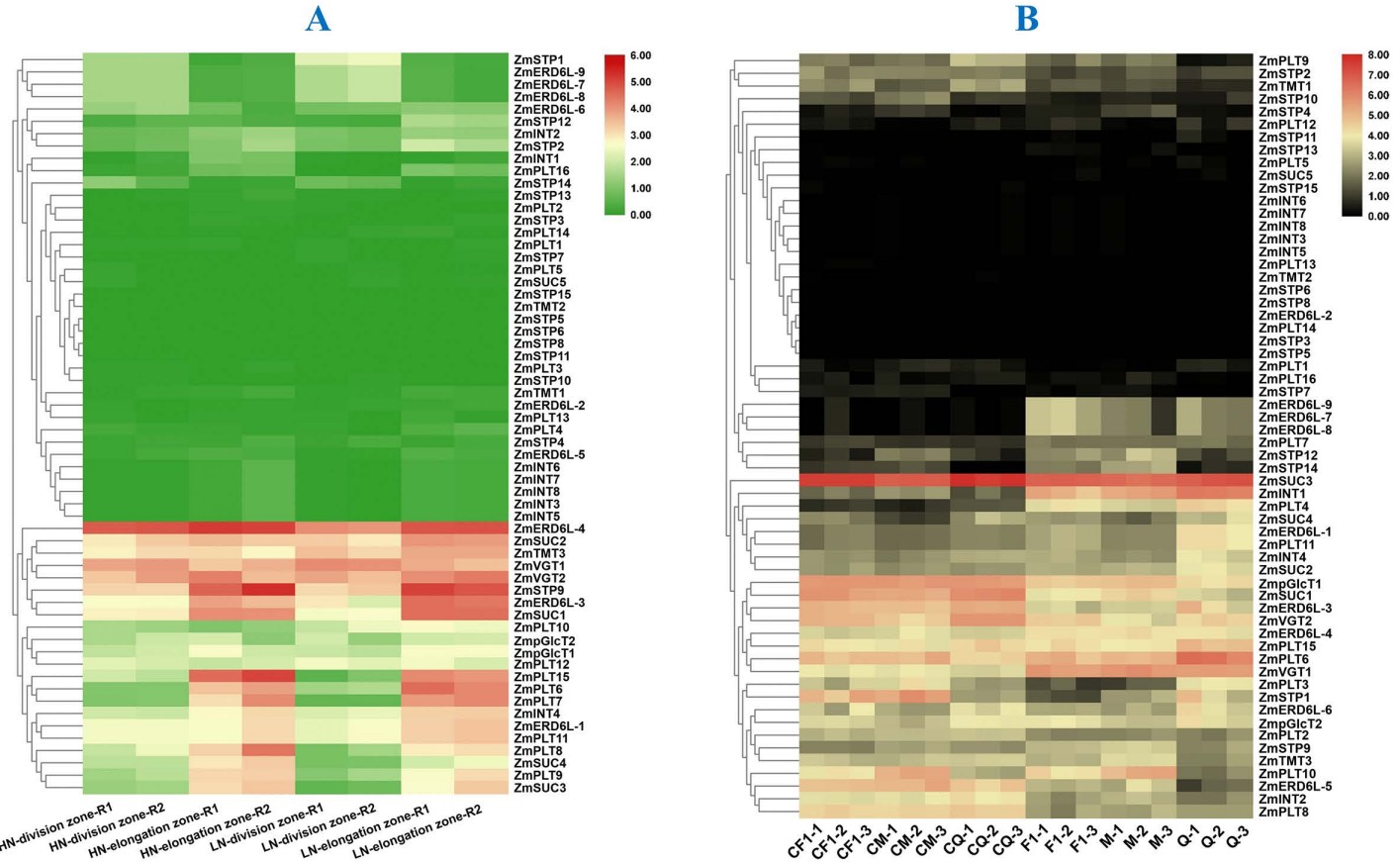

**Fig 9. Expression analysis of maize sugar transporters under nitrogen starvation (A) and heat stress condition (B). Note**: **PLT**: Polyol Transporter, **STP**: Sugar Transport protein, **ERD6**: Early Response to Dehydration 6, **INT**: Inositol Transporter, **SUC**: Sucrose Carrier, **TMT**: Tonoplastic Monosaccharide Transporter, **VGT**: Vacuolar Glucose Transporter, **pGlcT**: Plastidic Glucose Translocator. HN = High nitrogen, LN = Low nitrogen, F1 = First Filial Generation of Hybrid, M = Parental line of the hybrid, Q = Maternal line of hybrid, C = Control, non C = heat stress condition. 1,2,3 represent independent replications.

To validate the predicted findings of the bioinformatics analyses, the subcellular localization of ZmPLT1 and ZmPLT8 dual-luciferase reporter assays were performed to analyze the localization and observed by confocal microscopy. The results demonstrated that these transporters were localized in the plasma membrane (Fig 13), which is consistent with the bioinformatics-based result (Table 1). The consistent findings between wet and dry lab results enhance the confidence in the accuracy of the computational predictions, supporting their biological relevance in involvement of intercellular sugar transport.

## Discussion

Maize is a globally significant crop, providing a vital source of calories for approximately 4.5 billion people in developing nations, particularly in Africa and Mesoamerica [33]. Given its importance as a staple crop, understanding the mechanisms that regulate sugar transport in maize is crucial for improving yield and stress tolerance. Sugar transporters are essential in mediating the movement of sugars across plant tissues, and their roles in abiotic stress tolerance have been well documented [34]. Members of the sugar transporter gene family have been reported in several plant species [35–38]. However, the molecular mechanisms underlying the contribution of maize sugar transporters to total sugar accumulation

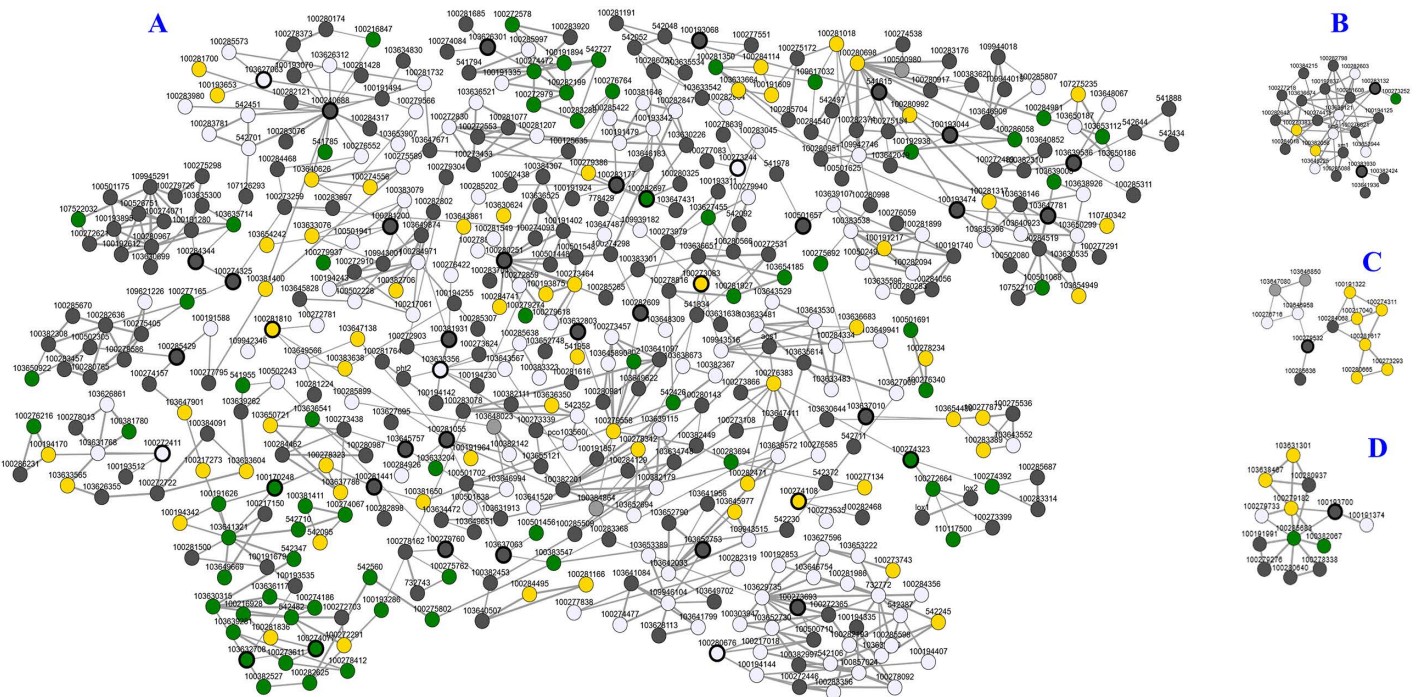

**Fig 10. Co-expression network of maize sugar transporters.** In the network, 491 genes clustered with 43 maize sugar transporter nodded in 4 co-expression clusters. Maize sugar transporters are marked in green, co-expressed genes in yellow, and others in white or grey. Cluster A includes genes related to stress response and nutrient transport, while clusters B, C, and D mainly show interactions with stress-associated and uncharacterized proteins.

and stress tolerance remain underexplored. This study aims to fill this gap by providing a comprehensive, integrative analysis of maize sugar transporters, combining genome-wide identification, expression analysis, and molecular dynamics simulations to elucidate their functional roles.

This study identified eight major sugar transporter subfamilies in maize, which provides a novel insights into their functional categorization and evolutionary relationships with the sugar transporters of other plant species [39]. Our findings are consistent with previous studies, which highlight the role of sugar STPs in sucrose and monosaccharide transport across developmental stages in plants, with *OsSTP11, 14,* and *20* are upregulated under heat stress, suggesting that *ZmSTPs* may similarly play a vital role in maize heat stress tolerance [39]. This insight demonstrates that *ZmSTPs* could be critical for maize adaptive responses to heat stress. Furthermore, *AtERD6Ls* play key roles in seed germination and stress responses in *Arabidopsis thaliana*, and similar functions might be conserved in maize. The *ZmERD6L* transporters suggests their importance in maize during development and stress adaptation [16]. *Arabidopsis* INTs are crucial for root development and stress tolerance suggest that maize INTs might have a great role in developing and tolerating abiotic stress [13]. Our findings further support *ZmINTs* as vital regulators in maize root development and stress adaptation. Our research identifies three additional sub-families of sugar transporters in maize SUC, TMT, and VGT which are crucial for growth, development, and stress tolerance [38,40]. The ZmpGlcTs found in maize are similar to pGlcT in olive trees, which helps move glucose (Glu) when starch breaks down and also plays a role in fruit ripening [41]. Most maize sugar transporters were localized to the plasma membrane, highlighting their role in maintaining cellular homeostasis and facilitating nutrient transport. Plasma membrane transporters are crucial for ion gradients and sugar mobilization, especially under stress, ensuring plant

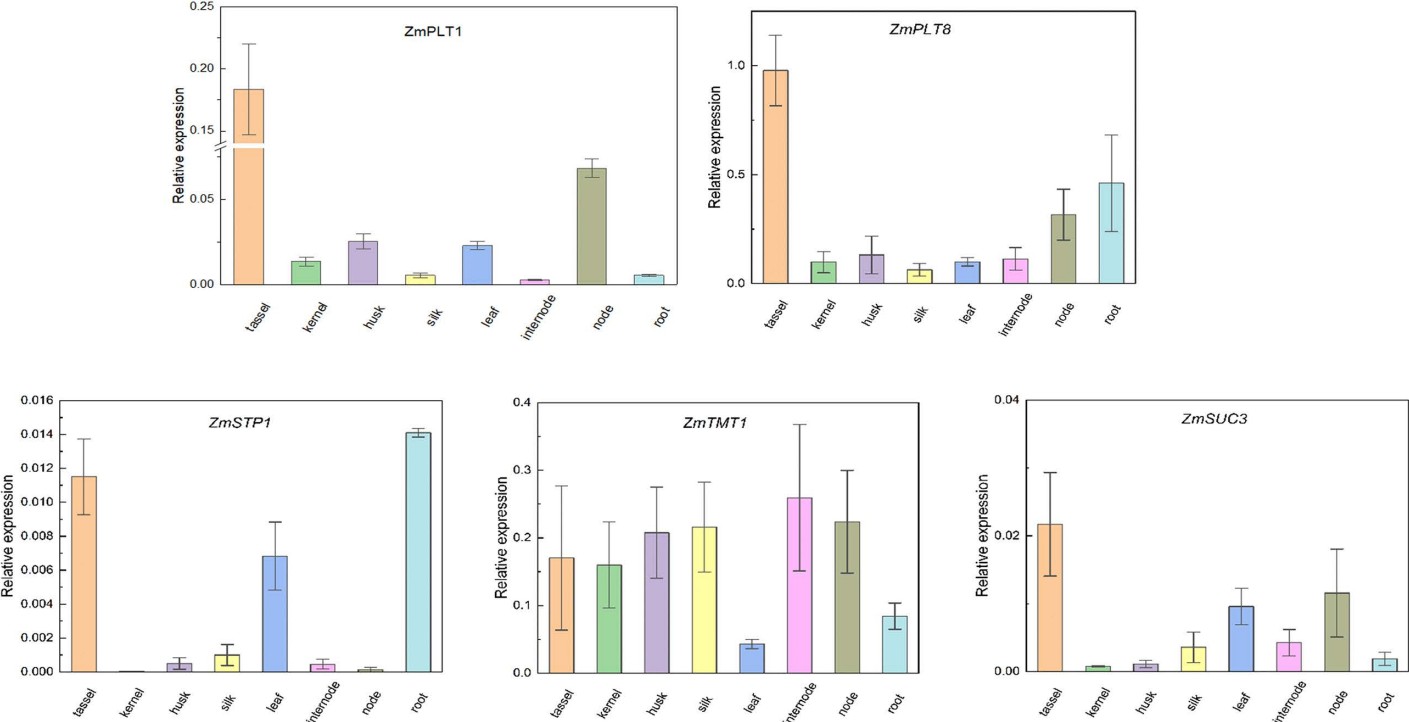

**Fig 11. Tissue-specific expressions of maize sugar transporters.** The qRT-PCR analysis was used to analyze the expression of selected sugar transporter genes in tassel, kernel, husk, silk, leaf, internode, node, and root. The names of the genes were labeled at the top of each diagram. Values were mean±standard error (SE) (n=3).

survival and productivity [42,43]. These findings emphasize the importance of maize sugar transporters in stress adaptation and resilience.

Phylogenetic analysis grouped 60 maize sugar transporters into eight clades, reflecting functional specificity and evolutionary conservation, consistent with previous studies [44]. Gene name inconsistencies with *Arabidopsis* and rice suggest gene duplication and divergence within maize [14]. These findings highlight the evolutionary dynamics of maize sugar transporters and set the stage for exploring their roles in stress tolerance and sugar mobilization.

The three-dimensional structures of sugar transporter proteins revealed a variety of coil topologies, transmembrane helices, and amphipathic alpha-helices, which are crucial for activating ion channels and facilitating the transport of signaling molecules across membranes [45,46]. These transmembrane helices play essential roles in signal transduction, nutrient transport, and protein trafficking, and are involved in various physiological processes, including interactions with disease pathways [47,48]. Transmembrane analysis revealed 11–12 helices in maize sugar transporters, with most having cytoplasmic N/C terminals which have extracellular N-terminal, suggesting a potential difference in the functional roles of these transporters in membrane-bound processes [49,50].

Synteny refers to the conservation of gene blocks across related species [51]. Our analysis showed that sugar transporters in maize and other plants share similar gene distributions, suggesting a common evolutionary origin [52]. The duplication of maize transporters in other species indicates their structural and functional conservation (Fig 3, S1 Table). Gene Ontology analysis highlighted the involvement of maize sugar transporters in various biological pathways, with similarities to rice sugar transporters linked to drought tolerance, suggesting a conserved role in stress adaptation [53,54].

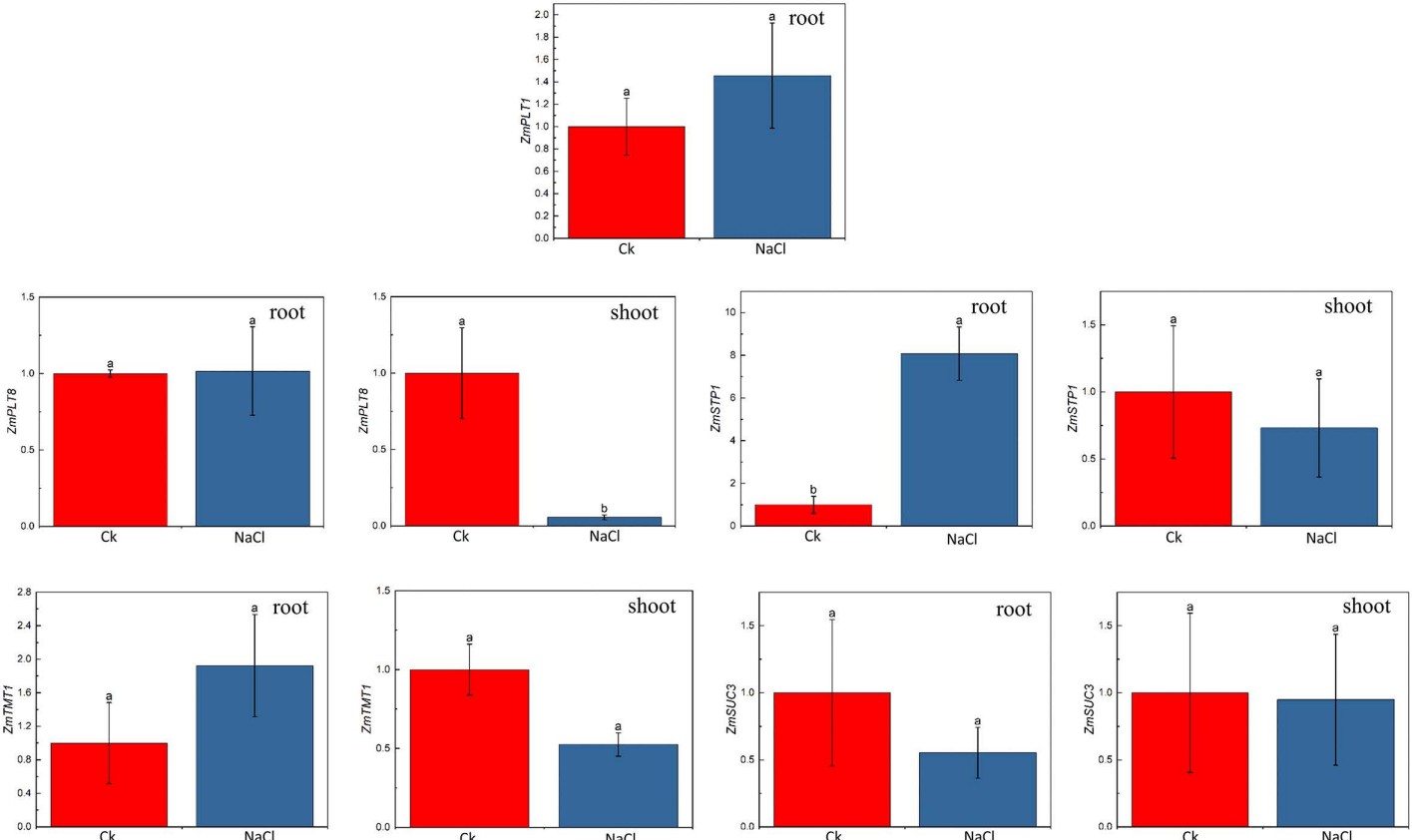

**Fig 12. qRT-PCR expression analysis of selected maize sugar transporters in root and shoot under salinity stress.** The error bars indicate the standard error of three replicates. The statistical significance of the difference in expression between the control and treated groups was analyzed. Data are means±SD of n=3 biological replicates, with different letters indicating significant differences in expression means at p<0.05.

Molecular docking is a vital technique for predicting ligand binding in structural biology [55]. Docking maize sugar transporters with four ligands revealed varying binding affinities, with similar results to those observed in maize SWEET transporters, suggesting a conserved mechanism of ligand binding across sugar transporter families [20]. Hydrogen and hydrophobic bonds were observed in maize sugar transporter-ligand complexes, highlighting the importance of both hydrogen and hydrophobic bonds in transporter specificity and efficiency. The interaction between Glu and polar residues around the pore channel of maize sugar transporter is crucial for sugar transport and substrate translocation [20]. Hydrophobic side chains further enhance bonding affinity and sugar transporter stability of maize sugar transporters [20]. Additionally, conserved Glu residues were identified in the docking results, which have been linked to protein stability in other transporters, suggesting potential targets for genetic improvement in maize sugar transporters [56]. This finding reinforces the role of conserved residues in maintaining transporter function and stability.

Significant interaction of 59 maize sugar transporters during PPI analysis mightsuggest functional roles in common metabolic pathways, with significant interactions linked to stress responses, growth, and biofuel production [57,58]. These findings are consistent with PPI regulation in rice, highlighting the importance of transporter proteins in stress tolerance and growth of plants [59]. Such interactions provide valuable insights for genetic improvements aimed at enhancing sugar accumulation and stress resilience in maize, underscoring the potential of PPI networks in crop optimization.

|  | GFP | Plasma membrane | Bright field | Merged |
|---|---|---|---|---|

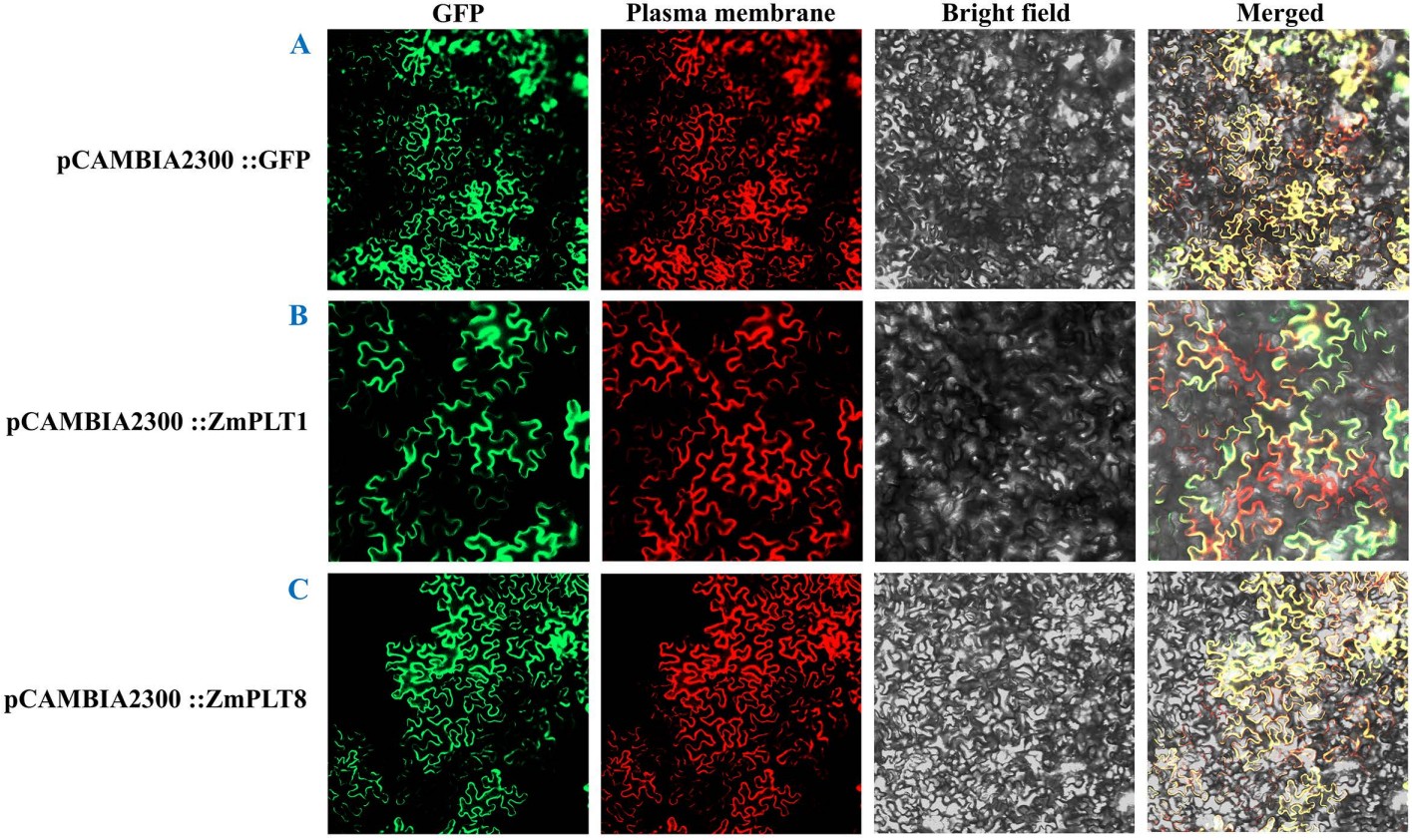

**Fig 13. Subcellular localization of ZmPLT1 and ZmPLT8.** The signals from the pCAMBIA2300::GFP **(A)**; ZmPLT1::GFP (B) and ZmPLT8::GFP **(C)**.

MD simulation analysis provides atomic-level insights into the movements of biomolecules, helping to understand ligand binding and protein conformational changes [60]. Our simulation results showed that RMSD values ranged from 1 to 3 Å, suggesting stable interactions and highlighting the importance of stable protein-ligand binding for transporter function. The RMSF analysis revealed flexibility at the N- and C-terminal regions, consistent with findings that these domains are crucial for substrate binding and transporter function [61,62]. Low Rg and high SASA values further highlight the compact and accessible structure of these complexes, which is essential for efficient sugar transport across membranes. These results suggest that maize sugar transporters play a critical role in sugar mobilization and stress adaptation. The computational findings lay a solid foundation for future experimental studies (genetic perturbations and in vivo assays) to validate their roles in sugar transport and stress tolerance in maize.

This study provides new insights into the role of sugar transporters in maize, particularly in relation to their involvement in stress tolerance. *ZmPLT4*, *ZmPLT8*, *ZmPLT10*, *ZmPLT11*, and *ZmPLT15* were highly expressed in the leaf, internode, primary root, and root cortex, indicating their crucial role in sugar loading in roots and unloading in leaves. In *Lotus japonicus*, PLTs are involved in sugar transport from source to sink, suggesting a similar function in maize [9]. The higher expression of *ZmSTP1, ZmSTP2,* and *ZmSTP10* in leaf and root tissues might suggest their potential involvement of sugar transport and accumulation in supporting energy distribution, growth, and stress responses in maize. *ZmTMT1* is highly expressed in all tissues, which play a key role in vacuolar sugar loading and regulation of sugar distribution for stress tolerance and proper metabolic balance in maize [14]. *ZmERD6L-4* showed moderate expression in all tissues, indicate its long-distance sugar transport in maize, to support energy distribution and stress adaptation.

The upregulation of these transporters under drought, salt, and heat, stress conditions might suggest their critical roles in maintaining sugar homeostasis during abiotic stress. *ZmSUC3* was particularly highly expressed in leaves and internodes, reflecting its key role in phloem loading and carbon distribution in maize [63]. Indeed, *ZmSTP1* was highly expressed in leaves under drought stress, while *ZmPLT1*, *ZmPLT3*, *ZmPLT4*, *ZmPLT6*, *ZmPLT8*, *ZmPLT10*, and *ZmPLT15* showed significant upregulation under salt stress. These findings are consistent with the role of sugar transporters in balancing metabolic processes and enhancing stress tolerance [64].

Co-expression networks reveal genes regulated by the same pathways, that provide deep insights into the biological processes [65]. In *Arabidopsis*, co-expression of stress-related transcription factors enhances the expression of *ERD1*, involved in dehydration stress [65]. Our study identified 491 genes clustered with 43 maize sugar transporters, which suggest their roles in regulating sugar transport and stress tolerance in maize.

Significant relative mRNA upregulation of *ZmPLT8*, *ZmSTP1*, *ZmSUC3*, *ZmPLT1*, and *ZmTMT1* in different tissues might be due to their great contribution in sugar transport and accumulation for regulating proper cellular homeostasis in expressed tissues (Fig 11). Under salt stress, significant upregulation of *ZmSTP1*, *ZmTMT1*, and *ZmPLT1* at the root might be due to their vital role in nutrient balance, sugar uptake, transport, and accumulation for maintaining cellular homeostasis under salinity stress. The possible functions of these transporters are supported by consistent results from qRT-PCR and *in silico* analysis, which show the highest degree of precision in experiments.

Predicted subcellular localization of ZmPLT1 and ZmPLT8 at the plasma membrane is consistent with the wet lab experiment (Fig 13). ZmSWEETs are plasma, vacuolar, and thylakoid membrane localized [20]. Membrane transporters have a great role in the intake of nutrients, hormone movement, allocation of resources, exclusion or sequestration of various solutes from cells, and environmental and developmental signaling. Hence, maize sugar transporters might play a significant role in cellular osmotic balance, vacuolar sugar storage, hormonal balance, and abiotic stress tolerance in maize. The research findings might help maize synthetic biologists create maize with improved sugar absorption, transport, accumulation, and abiotic stress tolerance by using a synthetic genetic circuit based on structural biology-guided programming.

## Conclusion

Maize sugar transporters play a vital role in long-distance sugar transport from source to sink. Following a series of bioinformatics analyses, the 60 identified sugar transporters in maize were found to be distributed into eight subfamilies. Among these, significant variations in 3D protein structure, membrane topology, and syntenic relationships, representing their substrate specificity and transport activity. Differential specificity of the proteins with sugar molecules during docking with different sugars might guide their substrate specificity with variations in transport affinity. MD simulation of ZmSTP14-fructose, ZmSTP10-galactose, ZmSTP10-glucose, and ZmVGT1-sucrose docked complexes that performed best in docking were performed. Strong Rg, hydrogen bonding in complexes, and constant fluctuations in RMSD and RMSF values among the complexes might guide how these transporters actively and significantly transport the aforementioned sugars. Additionally, significant upregulation of these transporters in various plant tissues, including source and sink tissues, under drought, salinity, nitrogen starvation, and heat stress conditions might guide their abiotic stress tolerance roles in addition to sugar transport. The qRT-PCR based validation and observation of upregulated expression of *ZmPLT1, 8, ZmSTP1, ZmTMT1,* and *ZmSUC3* genes under different tissue and salinity stress might reveal their important role in maintaining homeostasis, transporting sugar, and salinity stress tolerance. Experimental validation of predicted plasma membrane localized ZmPLT1 and ZmPLT8 proteins in the same might reveal their significant roles in cellular osmotic balance, and vacuolar sugar storage that might play potential roles in hormonal balance for increasing abiotic stress tolerance. Therefore, the findings of the current study might guide the exploration of the mechanism of how these transporters contribute to the total sugar accumulation and abiotic stress tolerance in maize interacting different signaling pathways. Further studies, including genetic knockout or overexpression lines are necessary to establish causal

relationships between these transporters and stress tolerance phenotypes. Experimental validation, such as functional assays in vivo, could be crucial to connect the observed expression patterns to specific stress tolerance mechanisms and to fully understand how these transporters functions at the molecular level.

## Supporting information

**S1 Table. The primers used in qRT-PCR.**
(XLSX)

**S2 Table. The detailed information on the syntenic relationship of the maize sugar transporter family proteins.**
(XLSX)

**S3 Table. The detailed information on the GO analysis of maize sugar transporters.**
(XLSX)

**S4 Table. Tissue-specific expression FPKM values of maize sugar transporters.**
(XLSX)

**S5 Table. Drought expression FPKM values of maize sugar transporters.**
(XLSX)

**S6 Table. Salinity expression FPKM values of maize sugar transporters.**
(XLSX)

**S7 Table. Nitrogen starvation expression FPKM values of maize sugar transporters.**
(XLSX)

**S8 Table. Heat expression FPKM values of maize sugar transporters.**
(XLSX)

**S9 Table. The detailed information on the co-expression analysis of maize sugar transporters.**
(XLSX)

**S1 Fig. Co-expression analysis of maize sugar transporters.**
(PPTX)

## Acknowledgments

We thank King Saud University, Riyadh, Saudi Arabia for providing technical supports in the molecular dynamics simulation experiments.

## Author contributions

**Conceptualization:** M. Atikur Rahman, Md. Mahmudul Hasan.

**Data curation:** Md. Sohel Mia, Rui Li, Fang Li.

**Formal analysis:** Md. Sohel Mia, Rui Li, Fang Li.

**Funding acquisition:** Tao Yang, M. Atikur Rahman, Md. Mahmudul Hasan.

**Investigation:** M. Atikur Rahman, Md. Mahmudul Hasan.

**Methodology:** M. Atikur Rahman, Md. Mahmudul Hasan.

**Supervision:** Tao Yang, Rui Li, Chao Xia, Tanveer A. Wani, M. Atikur Rahman, Md. Mahmudul Hasan.

**Writing – original draft:** Md Suzauddula.

**Writing – review & editing:** Md Suzauddula, Jianbo Mi, Seema Zargar, M. Atikur Rahman.

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
