## [Decision Letter · Decision Letter 0]

2 Jan 2026

Dear Dr. Hasan,

We look forward to receiving your revised manuscript.

Kind regards,

Mojtaba Kordrostami, Ph.D.

Academic Editor

PLOS One

**Journal Requirements:**

“The Natural Science Foundation of Sichuan Province (Grant No. 2022NSFSC1774), the National Natural Science Foundation of China (Grant No. 32101673), the Liangshan Prefecture Science and Technology Program (24YYYJ0184 and 24YYYJ0183), the Sichuan Science and Technology Program (Grant No. 2021YFYZ0017), and the Academician Expert Workstation of Yunnan Province (to Han Lan, Grant No. 202305AF150052) provided support for this work. The Yunnan Major special research project "Creation and application of special bio-fertilizer for Plateau characteristic economic crops in Yunnan Province" (202202AE090015) also provided money for the work. King Saud University in Riyadh, Saudi Arabia, (project number RSP2025R357) for providing technical support of molecular dynamics simulation experiment.”

5. Please note that funding information should not appear in any section or other areas of your manuscript. We will only publish funding information present in the Funding Statement section of the online submission form. Please remove any funding-related text from the manuscript.

6. "Please include a caption for figure 6.

7. Please include captions for your Supporting Information files at the end of your manuscript, and update any in-text citations to match accordingly. Please see our Supporting Information guidelines for more information: http://journals.plos.org/plosone/s/supporting-information .

Reviewers' comments:

Reviewer's Responses to Questions

**Comments to the Author**

1. Is the manuscript technically sound, and do the data support the conclusions?

Reviewer #1: Partly

Reviewer #2: Partly

2. Has the statistical analysis been performed appropriately and rigorously?

Reviewer #1: N/A

Reviewer #2: No

3. Have the authors made all data underlying the findings in their manuscript fully available?

Reviewer #1: Yes

Reviewer #2: No

4. Is the manuscript presented in an intelligible fashion and written in standard English?

Reviewer #1: No

Reviewer #2: No

Reviewer #1: This study represents a meaningful investigation into maize sugar transporters, which play crucial roles in transmembrane sugar transport and tissue-specific functions. The authors systematically identified eight subclasses of sugar transporters in maize, analyzed their sequence characteristics, and conducted expression profiling of ZmPLT1, ZmPLT8, ZmSTP1, ZmTMT1, and ZmSUC3 under various conditions. The membrane localization of ZmPLT1 and ZmPLT8 was experimentally validated. While the work provides foundational insights, substantial improvements are required for publication.

1. The manuscript requires thorough linguistic editing to resolve ambiguous phrasing. Notable examples include:

Abstract: "Co-expression of maize sugar transporter genes was co-expressed with stress responsive transcription factors MYB8-ZmSTP9 and A6b-ZmPLT10..." (ambiguous causal linkage)

Abstract: "After ZmPLT1, 8, ZmSTP1, ZmTMT1, and ZmSUC3 genes were validated by real-time PCR..." (redundant structure).

2. Line 49, Do the designations 'MYB8-ZmSTP9' and 'A6b-ZmPLT10' conform to the nomenclature conventions for transcription factors?

3. Line 50, “8” should revised to “ZmPLT8”.

4. Line 57, Reassess keyword selection to reflect core findings.

5. Line 58, Use standardized abbreviations: "qRT-PCR", "mL" (throughout).

6. Line 127, Ensure binomial nomenclature formatting (italicize Zea mays L. with roman "L").

7. Line 150, Specify database versions/access dates for PFAM, CDD, and SMART analyses.

8. Line 174, Report bootstrap values supporting phylogenetic tree topology.

9. Lines 245-248: Create dedicated paragraph for materials description.

10. Lines 300-304, Provide evolutionary significance rather than statistical counts.

11. Lines 336-339, Re-evaluate conclusions drawn from synteny analysis.

12. Line 487, Distinguish co-expression networks from protein-protein interactions.

13. Lines 281-285, Resolve numerical discrepancy: 59 vs. 60 reported genes.

14. Section 3.7: Remove redundant subheading hierarchy. Streamline bioinformatics narratives to enhance readability.

15. There are no bootstrap values in phylogenetic trees.

16. Methods: Describe protein interaction prediction methodology (missing for Fig. 5).

17. Fig. 5: Clarify methodology for protein interaction predictions.

18. Fig. 10: Replace numerical labels with interpretable biological annotations.

19. Fig. 13: Justify the "nucleus" designation in localization results, but the result is not correct.

20. Line 1082: Cite English-language version when available.

While the study contains valuable data, substantial reorganization and methodological clarifications are required to strengthen scientific rigor and narrative coherence. A focused revision addressing these concerns would significantly enhance the manuscript's impact.

Reviewer #2: I have reviewed the manuscript entitled “Exploring structural, functional, evolutionary, and genetic characteristics of sugar transporters in maize and their roles in abiotic stress tolerance” (PONE-D-25-21914), which presents a comprehensive in silico and experimental survey of sugar transporter gene families in maize, integrating phylogenetics, structural prediction, synteny, docking and molecular dynamics simulations, expression profiling, qRT-PCR validation, and subcellular localization. While the topic is relevant and the dataset assembled is extensive, the manuscript in its current form raises significant concerns regarding novelty, analytical depth, and suitability for publication in PLOS ONE. The study largely follows a well-established “genome-wide identification plus expression analysis” template that has been repeatedly applied to transporter and transcription factor families across many plant species, and the present work does not clearly articulate a conceptual or methodological advance beyond incremental expansion of descriptive analyses. The molecular docking and MD simulations, although technically executed, are not convincingly linked to biological function or experimental validation, and their inclusion appears speculative, with limited interpretive value for in vivo sugar transport or stress tolerance mechanisms. Similarly, the expression analyses under abiotic stresses are predominantly descriptive, lacking statistical testing, functional perturbation (e.g., mutant or overexpression lines), or causal evidence connecting specific transporters to stress tolerance phenotypes. The manuscript is also excessively long and diffuse, with extensive repetition of background information and methodological detail that obscures the central findings and weakens the overall narrative. Although the authors state that all relevant data are available within the manuscript and supplementary files, clearer organization and stronger synthesis would be required to make the work accessible to a broad readership. Overall, despite the breadth of analyses presented, the study does not sufficiently meet the standards of novelty, mechanistic insight, or analytical rigor expected for a general-interest journal such as PLOS ONE, and I therefore recommend rejection or, at minimum, major revision with a substantially strengthened focus on biological significance, reduced redundancy, and clearer justification of the study’s contribution beyond descriptive genomics.

**Do you want your identity to be public for this peer review?** For information about this choice, including consent withdrawal, please see our Privacy Policy

Reviewer #1: No

Reviewer #2: No

---

## [Author Response · Author response to Decision Letter 1]

26 Jan 2026

We sincerely appreciate the reviewers for their thorough evaluation and insightful comments, which have significantly improved the quality and scientific rigor of our manuscript. All revisions are done under track change option. In the revised version, all changes made in response to Reviewer 1 are highlighted in yellow, and the same for Reviewer 2 are highlighted in green for clarity and transparency. Below, we provide a detailed, point-by-point response to each comment raised by the reviewers. Additional revisions are under non-highlighted track change option.

In addition to revised version under track change option, we are also adding clean version of the manuscript.

Editorial office comment

Comment 1: Please include a figure label and title for Figure 1 to 13 in your main manuscript.

Response: Thank you for your comment. In revision, we have added the figure label to our revised manuscript. Please see the revised manuscript. Lines 986-1080 in the clean version and lines 1631-1723 in the track change version manuscript.

Response to the Journal requirements

Journal Requirements:

Comment 1: Please ensure that your manuscript meets PLOS ONE's style requirements, including those for file naming. The PLOS ONE style templates can be found at

Response: Thank you for your helpful comment. We have reviewed the PLOS ONE style guidelines, and we will ensure that the manuscript adheres to the required formatting, including file naming. We will follow the provided templates for both the manuscript body and the title/author affiliations section. The revised version of the manuscript will be submitted in full compliance with these formatting requirements.

Comment 2: Please note that PLOS ONE has specific guidelines on code sharing for submissions in which author-generated code underpins the findings in the manuscript. In these cases, we expect all author-generated code to be made available without restrictions upon publication of the work. Please review our guidelines at https://journals.plos.org/plosone/s/materials-and-software-sharing#loc-sharing-code and ensure that your code is shared in a way that follows best practice and facilitates reproducibility and reuse.

Response: Thank you for the comment. We would like to clarify that no author-generated or custom code was developed or used in this study. Therefore, the code-sharing policy is not applicable to our manuscript. All analyses were conducted using standard, widely available tools, and the methodology has been described in sufficient detail to ensure reproducibility.

Comment 3: We note that the grant information you provided in the ‘Funding Information’ and ‘Financial Disclosure’ sections do not match.

Response: Thank you for your comment. We have carefully reviewed the ‘Funding Information’ and ‘Financial Disclosure’ sections and corrected them to ensure that the grant numbers and funding details are accurate and consistent across both sections.

Comment 4: Thank you for stating the following financial disclosure:

“The Natural Science Foundation of Sichuan Province (Grant No. 2022NSFSC1774), the National Natural Science Foundation of China (Grant No. 32101673), the Liangshan Prefecture Science and Technology Program (24YYYJ0184 and 24YYYJ0183), the Sichuan Science and Technology Program (Grant No. 2021YFYZ0017), and the Academician Expert Workstation of Yunnan Province (to Han Lan, Grant No. 202305AF150052) provided support for this work. The Yunnan Major special research project "Creation and application of special bio-fertilizer for Plateau characteristic economic crops in Yunnan Province" (202202AE090015) also provided money for the work. King Saud University in Riyadh, Saudi Arabia, (project number RSP2026R357) for providing technical support of molecular dynamics simulation experiment.”

Response: Thank you for the comment. We would like to clarify that the funders were involved in specific aspects of the study. In particular, the funders contributed to the study design, formal analysis, technical support, and supervision of the research. The manuscript and cover letter have been amended to include the following statement:

“The funders were involved in study design, formal analysis, technical support, and supervision of the research. They had no role in data collection, decision to publish, or preparation of the manuscript.”

Please include our current funding status. Our corrected funding’s status is-

“The Natural Science Foundation of Sichuan Province (Grant No. 2022NSFSC1774), the National Natural Science Foundation of China (Grant No. 32101673), the Liangshan Prefecture Science and Technology Program (24YYYJ0184 and 24YYYJ0183), the Sichuan Science and Technology Program (Grant No. 2021YFYZ0017), and the Academician Expert Workstation of Yunnan Province (to Han Lan, Grant No. 202305AF150052) provided support for this work. The Yunnan Major special research project "Creation and application of special bio-fertilizer for Plateau characteristic economic crops in Yunnan Province" (202202AE090015) also provided money for the work. King Saud University in Riyadh, Saudi Arabia, (project number ORF-2026-357) for providing technical support of molecular dynamics simulation experiment.”

We appreciate the opportunity to clarify this information.

Comment 5: Please note that funding information should not appear in any section or other areas of your manuscript. We will only publish funding information present in the Funding Statement section of the online submission form. Please remove any funding-related text from the manuscript.

Response: Thank you for the comment. We have revised according to your guidance. Please see the revised manuscript.

Comment 6: "Please include a caption for figure 6.

Response: Thank you for the comment. Revised accordingly. Please see the revised figure.

Comment 7: Please include captions for your Supporting Information files at the end of your manuscript, and update any in-text citations to match accordingly. Please see our Supporting Information guidelines for more information: http://journals.plos.org/plosone/s/supporting-information.

Response: Thank you for the comment. Revised accordingly. Please see the revised manuscript.

Comment 8: If the reviewer comments include a recommendation to cite specific previously published works, please review and evaluate these publications to determine whether they are relevant and should be cited. There is no requirement to cite these works unless the editor has indicated otherwise.

Response: Thank you for the comment.

Reviewers' comments:

Reviewer's Responses to Questions

Comments to the Author

1. Is the manuscript technically sound, and do the data support the conclusions?

Reviewer #1: Partly

Reviewer #2: Partly

Response: We thank both reviewers for their careful evaluation of our manuscript. In response to their comments, we have thoroughly revised the manuscript to improve its technical soundness and to better demonstrate that the data support the conclusions. Specifically, we have clarified the experimental design, strengthened the description of controls and replication, and revised the Results and Discussion sections to ensure that all conclusions are directly supported by the presented data.

2. Has the statistical analysis been performed appropriately and rigorously?

Reviewer #1: N/A

Reviewer #2: No

Response: We thank the reviewer for this important comment. In response, we have carefully revised the statistical analysis to improve its rigor and clarity.

3. Have the authors made all data underlying the findings in their manuscript fully available?

Reviewer #1: Yes

Reviewer #2: No

Response: Thank you for the comment. We confirm that all data underlying the findings of this study have been made fully available without restriction in the revised manuscript.

4. Is the manuscript presented in an intelligible fashion and written in standard English?

Reviewer #1: No

Reviewer #2: No

Response: Thank you for the comment. The manuscript has been thoroughly revised to improve readability, clarity, and consistency with standard English. Grammatical errors, typographical issues, and ambiguous expressions have been corrected throughout the text.

5. Review Comments to the Author

Response to Reviewer 1

Reviewer #1: This study represents a meaningful investigation into maize sugar transporters, which play crucial roles in transmembrane sugar transport and tissue-specific functions. The authors systematically identified eight subclasses of sugar transporters in maize, analyzed their sequence characteristics, and conducted expression profiling of ZmPLT1, ZmPLT8, ZmSTP1, ZmTMT1, and ZmSUC3 under various conditions. The membrane localization of ZmPLT1 and ZmPLT8 was experimentally validated. While the work provides foundational insights, substantial improvements are required for publication.

Comment 1: The manuscript requires thorough linguistic editing to resolve ambiguous phrasing. Notable examples include:

Abstract: "Co-expression of maize sugar transporter genes was co-expressed with stress responsive transcription factors MYB8-ZmSTP9 and A6b-ZmPLT10..." (ambiguous causal linkage)

Abstract: "After ZmPLT1, 8, ZmSTP1, ZmTMT1, and ZmSUC3 genes were validated by real-time PCR..." (redundant structure).

Response: Thank you for the comment. We have revised the highlighted sentences to improve clarity, remove redundancy, and avoid unintended causal implications. Please see the revised manuscript.

2. Line 49, Do the designations 'MYB8-ZmSTP9' and 'A6b-ZmPLT10' conform to the nomenclature conventions for transcription factors?

Response: Thank you for the comment. We clarify that the terms “MYB8–ZmSTP9” and “A6b–ZmPLT10” were not intended to represent official transcription factor names. Rather, they were used to denote co-expression relationships between the transcription factors (MYB8 and A6b) and their associated target genes (ZmSTP9 and ZmPLT10), respectively. Please see the revised manuscript. Lines 57-63 in the clean version and lines 80-83 in the track change version.

3. Line 50, “8” should revised to “ZmPLT8”.

Response: Thank you for the comment. Revised accordingly, please see the revised manuscript. Line 62 in the clean version and line 89 in the track change version.

4. Line 57, Reassess keyword selection to reflect core findings.

Response: Thank you for the comment. Revised accordingly, please see the revised manuscript. Lines 71-72 in the clean version and lines 106-108 in the track change version.

5. Line 58, Use standardized abbreviations: "qRT-PCR", "mL" (throughout).

Response: Thank you for the comment. Revised accordingly, please see the revised manuscript. Lines 214 and 382 in the clean and track change version, respectively.

6. Line 127, Ensure binomial nomenclature formatting (italicize Zea mays L. with roman "L").

Response: Thank you for the comment. Revised accordingly, please see the revised manuscript. Line 101 and 199 in the clean and track change version, respectively.

7. Line 150, Specify database versions/access dates for PFAM, CDD, and SMART analyses.

Response: Thank you for the comment. Revised accordingly, please see the revised manuscript. Line 123 and 250 in the clean and track change version, respectively.

8. Line 174, Report bootstrap values supporting phylogenetic tree topology.

Response: We thank the reviewer for this comment. In the present study, the phylogenetic analysis was primarily conducted to classify maize, rice, and Arabidopsis sugar transporter proteins into distinct clades and to illustrate their overall evolutionary relationships. The major clades were consistently resolved and well supported by the maximum likelihood approach based on established sequence similarity and domain composition. Line 149 and 276 in the clean and track change version, respectively.

Given that the phylogenetic tree was not intended to infer fine-scale evolutionary divergence but rather to support gene family classification, bootstrap values were not included in the current analysis. We have clarified this purpose in the revised manuscript to avoid overinterpretation of the tree topology.

9. Lines 245-248: Create dedicated paragraph for materials description.

Response: Thank you for the comment. We have revised the manuscript to present the materials description in a dedicated paragraph. The plant material, growth conditions, sampling location, and sampling time points are now clearly described in a separate paragraph in the Materials and Methods section to improve clarity and organization. Lines 232-237 in the clean version and lines 365-370 in the track change version.

10. Lines 300-304, Provide evolutionary significance rather than statistical counts.

Response: Thank you for the comment. Revised accordingly, please see the revised manuscript. Lines 304-313 in the clean version and lines 449-463 in the track change version.

11. Lines 336-339, Re-evaluate conclusions drawn from synteny analysis.

Response: Thank you for the comment. Revised the manuscript accordingly. Please see the revised manuscript. Lines 352-357 in the clean version and lines 534-540 in the track change version.

12. Line 487, Distinguish co-expression networks from protein-protein interactions.

Response: Thank you for the comment. In the revised version, co-expression results are now explicitly described as indicators of potential functional association rather than physical interaction. Please see the revised manuscript. Line 488 and 803 in the clean and track change version, respectively.

13. Lines 281-285, Resolve numerical discrepancy: 59 vs. 60 reported genes.

Response: Thank you for the comment. Upon careful re-examination, we corrected th

---

## [Decision Letter · Decision Letter 1]

1 Feb 2026

Exploring structural, functional, evolutionary, and genetic characteristics of sugar transporters in maize and their roles in abiotic stress tolerance

PONE-D-25-21914R1

Dear Dr. Hasan,

We’re pleased to inform you that your manuscript has been judged scientifically suitable for publication and will be formally accepted for publication once it meets all outstanding technical requirements.

Kind regards,

Mojtaba Kordrostami, Ph.D.

Academic Editor

PLOS One

Additional Editor Comments (optional):

Reviewers' comments:

Reviewer's Responses to Questions

**Comments to the Author**

Reviewer #2: All comments have been addressed

2. Is the manuscript technically sound, and do the data support the conclusions?

Reviewer #2: Yes

3. Has the statistical analysis been performed appropriately and rigorously?

Reviewer #2: Yes

4. Have the authors made all data underlying the findings in their manuscript fully available?

Reviewer #2: (No Response)

5. Is the manuscript presented in an intelligible fashion and written in standard English?

Reviewer #2: Yes

Reviewer #2: Dear Editor

All comments have been adressed.

Thanks again for paying attention to my comments.

I can now advise accepting this manuscript

**Do you want your identity to be public for this peer review?** For information about this choice, including consent withdrawal, please see our Privacy Policy

Reviewer #2: No

---

## [Editor Report · Acceptance letter]

PONE-D-25-21914R1

PLOS One

Dear Dr. Hasan,

I'm pleased to inform you that your manuscript has been deemed suitable for publication in PLOS One. Congratulations! Your manuscript is now being handed over to our production team.

Kind regards,

on behalf of

Dr. Mojtaba Kordrostami

Academic Editor

PLOS One